# Rethinking the Temperature for Federated Heterogeneous Distillation

Fan Qi [1]   Daxu Shi [1]   Chuokun Xu [2]   Shuai Li [1]   Changsheng Xu [3]

## Abstract

Federated Distillation (FedKD) relies on lightweight knowledge carriers like logits for efficient client-server communication. Although logit-based methods have demonstrated promise in addressing statistical and architectural heterogeneity in federated learning (FL), current approaches remain constrained by suboptimal temperature calibration during knowledge fusion. To address these limitations, we propose **ReT-FHD**, a framework featuring: 1) Multi-level Elastic Temperature, which dynamically adjusts distillation intensities across model layers, achieving optimized knowledge transfer between heterogeneous local models; 2) Category-Aware Global Temperature Scaling that implements class-specific temperature calibration based on confidence distributions in global logits, enabling personalized distillation policies; 3) Z-Score Guard, a blockchain-verified validation mechanism mitigating 44% of label-flipping and model poisoning attacks. Evaluations across diverse benchmarks with varying model/data heterogeneity demonstrate that the ReT-FHD achieves significant accuracy improvements over baseline methods while substantially reducing communication costs compared to existing approaches. Our work establishes that properly calibrated logits can serve as self-sufficient carriers for building scalable and secure heterogeneous FL systems.

## 1 Introduction

In cross-device Federated Learning (FL), critical challenges arise from model heterogeneity (e.g., diverse architectures across devices), statistical heterogeneity (non-IID data distributions across clients), prohibitive communication costs, and privacy risks due to exposure of sensitive visual data. Integrating knowledge distillation (KD) into FL frameworks addresses these issues by enabling architecture-agnostic knowledge transfer from heterogeneous local models to a lightweight global model, minimizing direct data transmission, and preserving data privacy through indirect gradient-free knowledge aggregation.

Early federated knowledge distillation frameworks, for example, the FedKD (Jeong et al., 2018) transmit mean logit vectors from clients to a server. These vectors are averaged into global consensus outputs and redistributed to clients, where they serve as teacher signals for distillation regularization during local training. Addressing the model heterogeneity, FedMD (Li & Wang, 2019) proposes the clients communicate based on their output class scores on samples from the public dataset. Later, additional knowledge carriers are transferred for resolving *model heterogeneity*, such as features (He et al., 2020; Wang et al., 2024) and proxy models (Lee et al., 2024). However, transferring additional information, whether as proxy models (Wang et al., 2025) or synthetic data (Huang et al., 2024; Liang et al., 2024), introduces both security risks and significant computational and communication overhead.

In this paper, we investigate whether the minimal carrier of knowledge—logits—can effectively address model heterogeneity in FL without public datasets and additional information. Recently, multi-level distillation, i.e., applying knowledge distillation at intermediate layers, has been demonstrated to enhance both the comprehensiveness and coherence of knowledge transfer (Khan et al., 2024; Hao et al., 2024; Huang et al., 2025b). However, the temperature, which is a critical hyperparameter in distillation, is typically simplified to static or monotonically decreasing (Hao et al., 2024; Huang et al., 2025b). This temperature strategy fails to adequately address the intricacies of data and model heterogeneity in FL, significantly limiting the efficiency and flexibility of distillation across heterogeneous models.

Therefore, we rethink the temperature for distillation and propose **ReT-FHD** framework in federated heterogeneous learning from theoretically and empirically: 1) *Model Heterogeneity*: We incorporate the elastic temperature that as-

---

[1]School of Computer Science and Engineering, Tianjin University of Technology, Tianjin, China [2]School of Computer Science and Engineering, Southeast University, Nanjing, China [3]Institute of Automation, Chinese Academy of Sciences, Beijing, China. Correspondence to: Chuokun Xu <chuokunxu@seu.edu.cn>.

*Proceedings of the $42^{nd}$ International Conference on Machine Learning*, Vancouver, Canada. PMLR 267, 2025. Copyright 2025 by the author(s).

signs flexible temperatures to multi-level distillation. Theoretically, temperature scheduling balances staged logit entropy, enhancing distillation precision and inter-stage knowledge transfer. 2) *Data Heterogeneity*: We present global temperature scaling, which assigns category-specific temperatures, aligning with gradient disparity dynamics to mitigate class-wise update imbalances. Analysis shows this stabilizes convergence by harmonizing gradient variations. 3) *Security:* Integrated with blockchain-empowered FL (BC-FL), our framework employs Z-score verification to validate logit distributions against dynamic boundaries, enabling automated reward/punishment protocols that deter malicious behaviors. 4) Experiments across four benchmarks demonstrate ReT-FHD's superiority over centralized and decentralized (DeL) baselines.

## 2 Preliminaries

### 2.1 Federated Distillation

Federated Distillation (FD) research addresses three core challenges: 1) Data Heterogeneity: Orthogonal efforts focus on refining knowledge carriers for non-IID data. Itahara et al. (2021) mitigate statistical heterogeneity via entropy reduction aggregation, while Zhang et al. (2022) synthesize hard samples to amplify KD effectiveness in data-free FL. 2) Model Heterogeneity: Foundational works include FedMD (Li & Wang, 2019), which aligns logits from public datasets for consensus, and FedDF(Lin et al., 2020), which ensembles soft labels from unlabeled data. FedGKT (He et al., 2020) further bridges model capacity gaps by distilling compact client features into a large server model. 3) Security: Chang et al. (2019) minimizes leakage via robust mean estimation on public predictions, Gong et al. (2022) employs quantized noise to perturb shared logits.

To defend model poisoning attacks, Khan et al. (2024) propose HYDRA-FL by offloading some of the KD loss to a shallow layer via an auxiliary classifier for local-to-local distillation. In contrast, our framework transfers logits instead of model parameters, introducing additional challenges due to compounded heterogeneity in both data distributions and model architectures.

### 2.2 Model-Heterogeneous Federated Learning

For generalized scenarios of heterogeneous models, two mainstream methods are *Prototype-based methods* and *Proxy model methods*. The *prototype-based* framework primarily facilitates knowledge transfer through client collaboration in constructing a unified class prototype. Tan et al. (2022) propose FedProto, a prototype-based method that addresses heterogeneity challenges in FL. Clients communicate local abstract class prototypes, which the server aggregates into a global prototype, eliminating model parameter communication. Zhang et al. (2024a) propose FedTGP, a framework using Adaptive Boundary Contrastive Learning (ACL) to learn trainable global prototypes (TGP) on the server, improving prototype separability while maintaining semantic meaning. On the other hand, the *proxy model* method focuses on training a compact model at each participant as a "proxy" for the primary model. Shen et al. (2020) propose FML, a federated learning framework enabling joint training of a universal model and personalized models, allowing custom designs for diverse applications beyond standard FL limitations.Wu et al. (2022) propose FedKD, a federated learning framework enabling mutual learning of client student-teacher models and student model sharing for knowledge transfer, alongside a dynamic gradient approximation method to reduce communication costs. Moreover, Wang et al. (2025) propose FedType, a bidirectional knowledge distillation strategy to optimize private and proxy models jointly, along with an uncertainty-based behavioral imitation method to improve proxy model guidance.

While the aforementioned methods address the challenges of Heterogeneous Federated Learning, the resulting increase in communication and computational demands poses significant limitations to the practical deployment of federated systems.

### 2.3 Dynamic Temperature Knowledge Distillation

Dynamic temperature strategies in knowledge distillation (KD) can be broadly divided into two categories: *Global scheduling methods* and *Local adaptive methods*. Global scheduling methods adjust a single temperature to control overall distillation difficulty. Liu et al. (2022) propose to learn the temperature by performing meta-learning on the extra validation set. Li et al. (2022b) dynamically adjust a shared temperature via curriculum learning to progressively increase distillation difficulty from easy to hard. Local adaptive methods assign temperatures at finer granularity. Guo et al. (2023) adjust sample-wise temperatures based on teacher confidence, while Wei & Bai (2024) minimize sharpness differences between teacher and student logits to derive separate temperatures for each.

Although these methods emphasize the effect of temperature on the distillation process, they are not suitable for knowledge distillation of heterogeneous models in FL by multi-client non-iid data.

## 3 Proposed Method

### 3.1 Background

**Multi-level Knowledge Distillation**: Compared to the original Federated Distillation approach (Li & Wang, 2019), Hybrid Knowledge Distillation (Khan et al., 2024; Hao et al.,

2024) has been shown to significantly enhance robustness and accuracy in Federated Learning. This improvement is achieved by integrating an auxiliary knowledge distillation constraint within a shallow layer of the local model. The hybrid loss function for FL clients is formulated as following:

$$\mathcal{L} = \mathcal{L}_{\text{CE}}(y_c, y) + \sum_{l=1}^{L} \mathcal{L}_{\text{KL}}^{(l)}\left(\sigma\left(z_t^{(l)}, \tau\right), \sigma\left(z_s^{(l)}, \tau\right)\right) \tag{1}$$

Here, $\sigma$ denotes the softmax function, $z_s^{(l)}$ denotes the logits of the student model in layer $l$, $z_t^{(l)}$ denotes the logits of the teacher model in layer $l$. $\tau$ is called temperature and serve as the offsets of the softmax function, $\alpha$ and $\tau$ effectively adjusts the sharpness or softness of the probability distribution derived from the teacher's logits. The loss function contains two main elements:

- $\mathcal{L}_{\text{CE}}$ (Cross-Entropy Loss) is used to optimize the classification ability of the client model for good task performance.

- $\sum_{l=1}^{L} \mathcal{L}_{\text{KL}}^{(l)}$ (Multi-level Distillation Loss) is used to calculate the KL divergence between the teacher logits and the student logits at each level $l$ to ensure that knowledge at different levels is adequately transferred.

**Z-Score Distillation:** Sun et al. (2024) propose a logit Z-Score standardization as a pre-process to enable student to focus on essential logit relations from teacher rather than requiring a magnitude match. To bridge the capacity gap of heterogeneous clients, we rewrite softmax based on the mean $\bar{z}$ and variance $\text{Var}(\cdot)$ as follows:

$$\sigma(\mathbf{z}; \overline{\mathbf{z}}, \text{Var}(\mathbf{z}))^{(c)} = \frac{\exp\left(\mathcal{Z}(\mathbf{z}; \tau)^{(c)}\right)}{\sum_{n=1}^{\mathcal{N}} \exp\left(\mathcal{Z}(\mathbf{z}; \tau)^{(n)}\right)}, \tag{2}$$

$$s.t. \ \mathcal{Z}(\mathbf{z}; \tau)^{(c)} = \frac{z^{(c)} - \bar{z}}{\text{Var}(z)\tau}$$

where $z^{(c)}$ represents the $c$-th term of $z$, which is the $c$-th class specific logits value, and $\mathcal{Z}$ is the Z-score function.

### 3.2 Rethink the Temperature in Heterogenous Federated Distillation

Despite the effectiveness against poisoning attacks (Kundu et al., 2021; Khan et al., 2024), the temperatures $\tau$ in Eq.1 are fixed, which neglects the inherent data and model heterogeneity present in the distillation process. Typically, higher $\tau$ makes the distribution sharper (more confident), while lower $\tau$ makes it softer (less confident), thus influencing the gradient and the learning process of the student model. However, in heterogeneous federated distillation, both data and model heterogeneity necessitate finer temperature regulation to enhance the utilization effectiveness of global logits.

**Multi-level Elastic Temperature :** To address model heterogeneity, we dynamically adjust the temperature during knowledge distillation based on the hierarchical characteristics of the models, effectively bridging representation discrepancies across architectures at different levels. Our multi-level distillation framework employs stage-adaptive temperature scaling: each level's output features feed into the subsequent level, enabling dynamic temperature calibration based on the statistical divergence between successive stages. Specifically, if the Z-score difference between the logits of the next stage and the logits of the teacher's model is large, the temperature of the current stage should be increased to reduce the amount of knowledge in the logits of the current stage, and thus promote learning more information from the logits of the teacher's model. Conversely, when the difference in logits is small, lower the temperature so that the model focuses more on learning in other phases.

We rewrite $\tau_l$ in level $l$ as $\tau_l = G_l(\Delta Z), \Delta Z_l = \|Z_T - Z_{l+1}\|$. $Z_T$ is the $Z(\cdot)$ distribution of the global logits. $\Delta Z_l$ represents the difference in Z-Scores between the global logits and the local logits, which is crucial for adjusting the temperature of the distillation process. The modulation of the effect of $\Delta Z$ is achieved through the function $G$, defined as a logarithmic function, as follows:

$$G_l(\cdot) = \xi + \frac{\gamma \log(1 + \Delta Z_l)}{\log(1 + \Delta Z_{\max})}, \tag{3}$$

where $\gamma$ is a scaling factor that determines the sensitivity of the function, $\xi$ is the lower bound of the temperature adjustment represented by the initial value of the temperature, and $\Delta Z_{\max}$ is used as a normalisation factor to determine the sensitivity of the function.

**Category-Aware Global Temperature Scaling:** To address data heterogeneity, we calculate distinct temperatures for each category in the global logits, enabling flexible adaptation to diverse client data characteristics, enhancing robustness and inter-category fairness while mitigating challenges from heterogeneous data distributions. Here, we give the formula for adjusting the temperature of the $c$-th category of global logits:

$$\tilde{\tau}_c = \tilde{\tau}_c - \beta \cdot \frac{\mathcal{L}_{\text{CE}}(f(\tilde{x}_c)) - \frac{1}{C}\sum_{i=1}^{C} \mathcal{L}_{\text{CE}}(f(\tilde{x}_i))}{\max\left(\left|\mathcal{L}_{\text{CE}}(f(\tilde{x}_c)) - \frac{1}{C}\sum_{i=1}^{C} \mathcal{L}_{\text{CE}}(f(\tilde{x}_i))\right|\right)} \tag{4}$$

where $\beta$ controls the weight of the temperature update, $max(|.|)$ represents the absolute maximum value taken to control the stability of the update. Since cross entropy directly reflects the robustness of the category, it is chosen to regulate the temperature update.

**Algorithm 1** Algorithm of our **ReT-FHD**

**Input:** Dataset$\{D_1, D_2, \ldots, D_K\}$, $T$ ▷$T$: Maximum communication rounds
1: **Initialization:** $\mathcal{W} \leftarrow$ Worker, $\mathcal{V} \leftarrow$ Validator, $\mathcal{M} \leftarrow$ Miner, Global logits $z^g$, Blacklist $\mathcal{B}$, Client neighbors set $\mathcal{N}$, Honest server $\mathcal{S}$, Client reward set $\mathcal{R}$
2: **for** $t = 1 \rightarrow T$ **do**
3:    each $k \in \mathcal{K} \leftarrow Randomly\{\mathcal{W}, \mathcal{V}, \mathcal{M}\}$ , each $k \in \mathcal{K} \leftarrow \{\mathcal{W}\}$
4:    **for** each $w \in [\mathcal{W}]$ **do**
5:      $w$ downloads $z_{t-1}^g$ from blockchain → **do** local update with Eq.6
6:      →sends $z_t^w$ to $v \in \mathcal{V}$
7:      $w$ downloads $z_{t-1}^g$ from $\mathcal{S}$ and **do** local update with Eq.5→ sends $z_t^w$ to $\mathcal{S}$
8:      $w$ gathers $z_{t-1}^n$ from $n \in \mathcal{N}_w$
9:    **end for**
10:    **for** each $v \in \mathcal{V}$ or $w \in \mathcal{W}$ or $\mathcal{S}$ **do**
11:      $v$ evaluates $w \in \mathcal{W}$ with $z_{t-1}^g$ by Eq.7
12:      → Auto set $r$ for each $w \in \mathcal{W}$ by Eq.8
13:      $v$ votes $w \in \mathcal{W}$ → sends to $m \in \mathcal{M}$
14:      $\mathcal{S}$ evaluates $w \in \mathcal{W}$ by Eq.7
15:      → set $r_k$ for each $w$ by Eq.8 → gather $z_{t+1}$ by $r_k \in \mathcal{R}$
16:      $w$ evaluates $z_{t-1}^n$ by Eq.7 → set $r_w$ for each $n \in \mathcal{N}_w$ by Eq.8
17:      → gather $z_t^w$ by $r_i^w \in \mathcal{R}^w$
18:      → **do** local update with Eq.6→ sends $z_t^w$ to $n \in \mathcal{N}_w$
19:    **end for**
20:    $m \in \mathcal{M}$ aggregates $z_t$ according to the weight of R, makes $\mathcal{B}$ by $\sum_{i=1}^v \mathcal{V}^o$
21:    → mines a block → uploads $\mathcal{B}$ , $R$ , $z_t^g$ to blockchain
22: **end for**

---

At the beginning of each training round, the client receives the global logits and then first adjusts the global logits according to the temperature $\tilde{\tau}$ calculated at the end of the previous round.

**Local update:** Ultimately, the FL optimization function can be expressed as:

$$\min_w \sum_{i=1}^{K} \frac{|\mathcal{D}_i|}{|\mathcal{D}|} \mathbb{E} \left[ \mathcal{L}_{\text{CE}} + \sum_{l=1}^{L} \sum_{c=1}^{C} \mathcal{L}_{\text{KL}}^{(l)} \left( \sigma^t(\cdot), \sigma^s(\cdot) \right) \right]$$
$$s.t. \; \sigma^t(\cdot) \sim \sigma\left(\mathbf{z_t}; \bar{z}_t, \text{Var}(\mathbf{z}_t), G(\Delta Z), \tilde{\tau}_c\right),$$
$$\sigma^s(\cdot) \sim \sigma\left(\boldsymbol{z_s}; \bar{z}_s, \text{Var}(\mathbf{z}_s), G(\Delta Z), \cdot\right)$$
$$(5)$$

where $\tilde{\tau}_c$ is the isolated temperature for category $c$ in the global logits.

The DeL optimization function can be expressed as:

$$\theta_i^{t+1} = \theta_i^{t+\frac{1}{2}} - \sum_{j \in \mathcal{N}(i)} \sum_{l=1}^{L} \mathcal{L}_{\text{KL}}^{(l)} \left( \sigma^t(\cdot), \sigma^s(\cdot) \right)$$
$$s.t. \; \sigma^t(\cdot) \sim \sigma\left(\mathbf{z_t}; \bar{z}_t, \text{Var}(\mathbf{z}_t), G(\Delta Z), \tilde{\tau}\right),$$
$$\sigma^s(\cdot) \sim \sigma\left(\boldsymbol{z_s}; \bar{z}_s, \text{Var}(\mathbf{z}_s), G(\Delta Z), \cdot\right)$$
$$(6)$$

## 3.3 Different FL Modes of Our ReT-FHD

The ReT-FHD framework supports distributed training through multilevel elastic distillation and statistical logit analysis, enabling intrinsic security and scalable FL deployment.

**FL and DeL mode:** In the Algorithm 1, the situation will not be so complicated for federated or decentralized learning. Each node belongs to the $w \in \mathcal{W}$. For FL mode, each client computes a multilayer distillation loss function with the server-side global logits, highlighted in yellow. For DeL mode, multilayer distillation training is performed after verifying the legitimacy of neighboring node logits, highlighted in pink.

**BC-FL mode:** Following the blockchain FL(Chen et al., 2021; Ying et al., 2024), we apply the POS voting-based consensus mechanism, highlighted in purple of Algorithm 1. In each round $t$, each $k \in \mathcal{K}$ is randomly assigned a role, including $w \in \mathcal{W}$,$v \in \mathcal{V}$,$m \in \mathcal{M}$, where $|\mathcal{W}|+|\mathcal{V}|+|\mathcal{M}| = |\mathcal{K}|$. $w$ is responsible for performing computations related to $f_i^w$. Each $w$ download global logtis $z_i$ from the latest block and train with private local data $D_i$. At this point the $w$ node performs the $E$ round local update and broadcasts the logits $z_i^w$, corresponding to the number of dimensions of the dataset categories, to the $v \in \mathcal{V}$ in the network. See the Appendix for more details.

**Z-Score Guard:** *Standardized Finding of Malicious Logits:* $\mathcal{Z}(\cdot, \cdot)$ plays a crucial role in evaluating logits for heterogeneous client models. This property likewise suggests that $\mathcal{Z}(\cdot, \cdot)$ can also play an important role in blockchain authentication sessions. In the verification process, validators $\mathcal{V}$ (verifiers) train locally and upload logits, while examinees $\mathcal{E}$(validators) evaluate $\mathcal{V}$'s logits. Under data heterogeneity, label sets across nodes may partially overlap or diverge. We define the labeling relationship among nodes as: $|\mathcal{V} \cap \mathcal{H}| = \eta$, $|\mathcal{E} \cap \mathcal{H}| = \phi$, and $|\mathcal{V} \cap \mathcal{E}| = \psi$. Here, $\mathcal{H}$ represents the complete set of labels for a distributed classification task.

The importance of $\psi$, $\eta$ and $\phi$ arises particularly if $\mathcal{E}$ acts as a lazy node by re-uploading $\mathcal{Z}(z_g, \cdot)$, which impedes the verification process by $\mathcal{V}$. To facilitate this verification, we suggest plotting both $\mathcal{Z}(z_g, \tau)^{(\psi)}$ and $\mathcal{Z}(z_\mathcal{V}, \tau)^{(\psi)}$ within a coordinate system:

$$\mathcal{D}_\mathcal{E} = \frac{\sum_{i=1}^{H} \left[ \exp\left(\left(\Delta Z_{\mathcal{E},\mathcal{V}}^i\right)^\psi\right) + \exp\left(\left(\Delta Z_{\mathcal{E},g}^i\right)^\phi\right) \right]}{\sum_{i=1}^{H} \exp\left(\left(\Delta Z_{\mathcal{V},g}^i\right)^\eta\right)}$$
$$> 2 \sum_{i=1}^{H} \exp\left( \mathcal{Z}\left(z_\mathcal{E}^i, \tau\right)^{(\psi)} - \frac{\mathcal{Z}\left(z_g^i, \tau\right)^{(\psi)} + \mathcal{Z}\left(z_\mathcal{V}^i, \tau\right)^{(\psi)}}{2} \right)$$
$$s.t. \; \Delta Z_{a,b}^i = \mathcal{Z}\left(z_a^i, \tau\right) - \mathcal{Z}\left(z_b^i, \tau\right)$$
$$(7)$$

Here $\tau$ is a fixed temperature. This approach allows us to quantify the criteria necessary for a verified individual to be considered valid or to be subject to voting, based on their positional relationship within the coordinate system. When $Z(z_{\mathcal{E}}, \cdot)$ equilibrium between $Z(z_{\mathcal{V}}, \cdot)$ and $Z(z_g, \cdot)$ is reached, $v$ will evaluates highest scores and vote bad.

*Logits-based Secure Validation:* Logits, as a crucial medium for information transmission, can also indicate the confidence level of a sample, in particular, the confidence level can demonstrate the knowledge of the model and also reveal information about the security of the model, which can be solved by looking at the logits relationship in the distributed system. We dynamically assign rewards based on this confidence. Where $R_b$ is the base reward, $\|D_i\|$ represents the number of samples per client, $\lambda$ is the preset confidence range, we provide a base algorithm for measuring reward values as follows:

$$\mathcal{R} = \begin{cases} R_b \cdot \|D_i\| \cdot \max\{P(y=i \mid x)\}, \text{if } \max\{P(\cdot)\} > \lambda \\ R_b \cdot \|D_i\| \cdot (1 - \max\{P(y=i \mid x)\}), \text{otherwise} \end{cases} \tag{8}$$

## 4 Theoretical Analysis

In this section, we systematically demonstrate the effectiveness of the proposed Multi-level Elastic Temperature and Global Temperature Scaling mechanism from the perspectives of information entropy and gradient updating.

Referring to Zhao et al. (2024), we introduce the knowledge scale to quantify the level of knowledge of the predictive distribution, which can measure the scale of knowledge learnt from the probability distribution by the model with randomly initialized weights:

$$K^T = KL(I(x), T(x)) = \log C - H(P^T) \tag{9}$$

where $KL(\cdot)$ stands for Kullback–Leibler divergence, which calculates the cost of information required to travel from one distribution to another, $C$ is the total number of categories and is a constant for the same dataset, and $H(\cdot)$ is the information entropy.

From the definition of knowledge scale, it can be found that the information entropy can be directly used to measure the knowledge level of teachers. Therefore, we obtain the following theorem by studying the partial derivative of information entropy with respect to temperature.

**Theorem 4.1.** *Temperature $\tau$ is positively correlated with the information entropy of the predictive distribution of logits.*

The proof of Theorem 4.1 can be found in Appendix A.1. Therefore, combining Theorem 4.1 and the definition of knowledge scale, the larger the temperature, the smaller the

knowledge scale, and the more local logits can learn from the global logits that can be learned.

In order to alleviate the problem of uneven distribution of category information due to data heterogeneity, we further introduce Global Logits Scaling, but before that, we need to discuss the effect of the previous Multi-level Elastic Temperature on categories.

**Corollary 4.2.** *For a model $f$ with random distribution $I$ in the initial state, for a deep neural network (DNN) model $f$ with random distribution $I$ in the initial state, the model has no preference for any sample. Thus for the model's predictive distribution $f(x; \theta_I) = \{p_1^I, p_2^I, ..., p_c^I\}$, the following relationship exists:*

$$\mathbb{E}(p^I(x)) = \mathbb{E}(p_1^I(x)) = \mathbb{E}(p_2^I(x)) = ... = \mathbb{E}(p_c^I(x)) \tag{10}$$

*We assume that the model is optimised under the guidance of a soft label distribution with smoothing degree $\lambda_1$, which is satisfied:*

$$p_1^{\lambda_1}(x_1) = p_2^{\lambda_1}(x_2) = \cdots = p_c^{\lambda_1}(x_c) > \frac{1}{C} >$$
$$p_1^{\lambda_1}(x_{c \neq 1}) = p_2^{\lambda_1}(x_{c \neq 2}) = \cdots = p_c^{\lambda_1}(x_{c \neq c}) \tag{11}$$

*The following relationship exists for the partial derivatives of the knowledge scale for different classes of logits:*

$$\frac{\partial KL(f(x; \theta_I), P_{\lambda_1})}{\partial z_1(x)} = \cdots = \frac{\partial KL(f(x; \theta_I), P_{\lambda_1})}{\partial z_c(x)} \tag{12}$$

Corollary 4.2 shows that the overall change in knowledge scale has a balanced impact on different categories. The proof of Corollary 4.2 can be found in Appendix A.2.

Now we can discuss the theory of Global Logits Scaling. Here, we give Theorem 4.3 about the relationship between the category temperature of global logits and the category gradient update of local models.

**Theorem 4.3.** *Following the setup in Corollary 4.2, the temperature $\tilde{\tau}_c$ of category $c$ is positively correlated with the gradient update $\nabla_{f \sim \lambda}^c$ of the local model category $c$.*

The proof of Theorem 4.3 can be found in Appendix A.3. Theorem 4.3 states that different temperatures affect the gradient update. Increasing the temperature of category $c$ will cause the gradient update $\nabla_{f \sim \lambda}^c$ of that category to increase, and vice versa will cause the gradient update $\nabla_{f \sim \lambda}^c$ of the category to decrease, and by regulating the temperature of different categories, it will cause the gradient update of different categories to be more balanced.

*Table 1.* Testing accuracy (%) in FL model heterogeneous ($a.$) and homogeneous ($b.$) settings. *Dir-set*: Dirichlet distribution; *Pat-set*: Pathological distribution.

| | Methods | | Tiny ImageNet | | Flower102 | | Cifar-100 | | Cifar-10 | |
|---|---|---|---|---|---|---|---|---|---|---|
| | | | *Dir-set* | *Pat-set* | *Dir-set* | *Pat-set* | *Dir-set* | *Pat-set* | *Dir-set* | *Pat-set* |
| ($a.$) | Proto-based | FedProto | $14.68_{\pm2.61}$ | $24.91_{\pm1.38}$ | $21.68_{\pm1.65}$ | $55.12_{\pm1.73}$ | $21.71_{\pm3.03}$ | $32.00_{\pm2.48}$ | $60.48_{\pm1.32}$ | $75.89_{\pm2.29}$ |
| | | FedTGP | $14.94_{\pm0.51}$ | $40.56_{\pm1.42}$ | $27.15_{\pm1.46}$ | $68.76_{\pm1.42}$ | $26.24_{\pm0.72}$ | $62.08_{\pm1.37}$ | $68.08_{\pm1.06}$ | $89.00_{\pm1.26}$ |
| | | RethinkFL | $15.21_{\pm1.65}$ | $38.47_{\pm0.80}$ | $30.36_{\pm1.19}$ | $65.24_{\pm1.42}$ | $28.37_{\pm0.39}$ | $63.11_{\pm2.44}$ | $68.21_{\pm2.16}$ | $88.96_{\pm3.24}$ |
| | Proxy-models | FML | $14.59_{\pm0.54}$ | $35.74_{\pm1.13}$ | $21.63_{\pm1.07}$ | $52.27_{\pm1.31}$ | $21.20_{\pm0.48}$ | $49.55_{\pm1.12}$ | $61.57_{\pm1.07}$ | $82.41_{\pm0.56}$ |
| | | FedKD | $14.57_{\pm0.65}$ | $32.07_{\pm1.01}$ | $20.76_{\pm1.27}$ | $52.71_{\pm1.43}$ | $19.48_{\pm0.58}$ | $51.58_{\pm1.13}$ | $63.32_{\pm1.06}$ | $83.73_{\pm0.51}$ |
| | | FedType | $16.05_{\pm0.64}$ | $38.84_{\pm0.92}$ | $23.53_{\pm0.86}$ | $60.89_{\pm1.24}$ | $24.21_{\pm0.31}$ | $52.51_{\pm1.34}$ | $66.43_{\pm1.23}$ | $84.28_{\pm0.52}$ |
| | | HYDRA-FL | $14.56_{\pm1.62}$ | $30.87_{\pm1.09}$ | $22.27_{\pm2.69}$ | $55.18_{\pm2.24}$ | $20.82_{\pm1.94}$ | $48.50_{\pm0.92}$ | $61.74_{\pm2.71}$ | $79.37_{\pm1.75}$ |
| | Logits-based | FedDistill | $15.20_{\pm0.62}$ | $37.67_{\pm1.28}$ | $20.47_{\pm1.31}$ | $62.27_{\pm1.58}$ | $21.04_{\pm0.57}$ | $53.72_{\pm1.17}$ | $63.51_{\pm1.03}$ | $83.08_{\pm0.64}$ |
| | | FCCL | $15.49_{\pm0.27}$ | $38.01_{\pm0.46}$ | $25.20_{\pm0.43}$ | $65.61_{\pm1.17}$ | $22.72_{\pm0.56}$ | $58.62_{\pm0.70}$ | $64.29_{\pm1.37}$ | $86.58_{\pm0.83}$ |
| | | **ReT-FHD(Ours, FL)** | $\mathbf{17.24_{\pm0.53}}$ | $\mathbf{42.35_{\pm0.98}}$ | $\mathbf{33.16_{\pm1.46}}$ | $\mathbf{72.91_{\pm0.97}}$ | $\mathbf{30.77_{\pm0.47}}$ | $\mathbf{68.56_{\pm0.66}}$ | $\mathbf{72.76_{\pm0.77}}$ | $\mathbf{91.23_{\pm0.65}}$ |
| ($b.$) | Proto-based | FedProto | $31.48_{\pm1.64}$ | $37.12_{\pm0.78}$ | $53.14_{\pm1.02}$ | $66.35_{\pm1.24}$ | $48.01_{\pm0.49}$ | $63.49_{\pm0.58}$ | $86.32_{\pm1.26}$ | $85.04_{\pm0.46}$ |
| | | FedTGP | $34.08_{\pm1.69}$ | $39.92_{\pm0.77}$ | $58.27_{\pm0.90}$ | $72.66_{\pm1.12}$ | $49.32_{\pm0.83}$ | $64.46_{\pm0.66}$ | $88.38_{\pm1.21}$ | $89.10_{\pm0.51}$ |
| | | RethinkFL | $35.04_{\pm2.72}$ | $37.50_{\pm1.11}$ | $60.39_{\pm0.94}$ | $67.95_{\pm1.52}$ | $48.34_{\pm0.26}$ | $63.90_{\pm1.47}$ | $89.33_{\pm1.06}$ | $88.35_{\pm1.94}$ |
| | Proxy-models | FML | $34.13_{\pm1.49}$ | $37.50_{\pm0.79}$ | $53.58_{\pm1.04}$ | $66.21_{\pm1.04}$ | $45.61_{\pm0.56}$ | $60.68_{\pm0.49}$ | $87.56_{\pm1.14}$ | $87.26_{\pm0.50}$ |
| | | FedKD | $33.99_{\pm1.56}$ | $37.97_{\pm0.77}$ | $53.82_{\pm1.05}$ | $66.06_{\pm1.05}$ | $48.62_{\pm0.42}$ | $63.98_{\pm0.46}$ | $88.38_{\pm1.10}$ | $89.05_{\pm0.53}$ |
| | | FedType | $34.29_{\pm0.69}$ | $40.74_{\pm1.06}$ | $60.12_{\pm1.15}$ | $75.27_{\pm2.03}$ | $47.38_{\pm0.48}$ | $64.73_{\pm0.93}$ | $89.77_{\pm1.82}$ | $88.91_{\pm0.76}$ |
| | | HYDRA-FL | $34.45_{\pm0.88}$ | $44.62_{\pm0.73}$ | $58.27_{\pm1.59}$ | $70.63_{\pm1.11}$ | $51.18_{\pm0.90}$ | $59.71_{\pm1.52}$ | $87.04_{\pm0.85}$ | $85.92_{\pm1.06}$ |
| | Logits-based | FedDistill | $33.71_{\pm1.58}$ | $37.73_{\pm0.80}$ | $55.67_{\pm0.95}$ | $65.37_{\pm1.05}$ | $49.39_{\pm0.76}$ | $63.51_{\pm0.48}$ | $88.56_{\pm1.13}$ | $88.19_{\pm0.50}$ |
| | | FCCL | $33.95_{\pm0.94}$ | $38.36_{\pm1.83}$ | $56.77_{\pm0.49}$ | $69.75_{\pm1.15}$ | $51.52_{\pm0.78}$ | $64.42_{\pm0.57}$ | $88.29_{\pm1.45}$ | $87.98_{\pm0.16}$ |
| | | **ReT-FHD(Ours, FL)** | $\mathbf{35.43_{\pm1.52}}$ | $\mathbf{50.99_{\pm0.48}}$ | $\mathbf{61.56_{\pm0.98}}$ | $\mathbf{89.66_{\pm1.01}}$ | $\mathbf{51.97_{\pm0.72}}$ | $\mathbf{68.49_{\pm0.46}}$ | $\mathbf{90.54_{\pm1.03}}$ | $\mathbf{89.85_{\pm0.57}}$ |

*Table 2.* Compared to decentralized methods in four datasets.

| Method | Cif-10 | Cif-100 | Flow. | Tiny. |
|---|---|---|---|---|
| Local SGD | $87.50_{\pm1.37}$ | $55.47_{\pm2.08}$ | $62.43_{\pm1.13}$ | $35.26_{\pm2.60}$ |
| DPSGD | $83.01_{\pm1.31}$ | $40.56_{\pm1.24}$ | $50.11_{\pm1.58}$ | $23.57_{\pm1.95}$ |
| CGA | $65.65_{\pm2.37}$ | $30.81_{\pm3.82}$ | $35.62_{\pm2.86}$ | $18.85_{\pm3.88}$ |
| Coll-FL | $87.14_{\pm5.28}$ | $50.65_{\pm3.52}$ | $64.21_{\pm4.66}$ | $37.45_{\pm3.59}$ |
| L2C | $90.14_{\pm0.34}$ | $59.00_{\pm0.42}$ | $65.92_{\pm0.71}$ | $37.95_{\pm0.58}$ |
| DeSA | $84.14_{\pm0.51}$ | $60.12_{\pm0.89}$ | $66.68_{\pm1.64}$ | $38.05_{\pm0.37}$ |
| **ReT-FHD(Ours, Del)** | $88.80_{\pm5.07}$ | $64.11_{\pm5.51}$ | $67.46_{\pm5.16}$ | $38.21_{\pm5.04}$ |

# 5 Experiment

## 5.1 Experimental Setup

**Datasets and Heterogeneity Setting**
*Datasets:* Cifar10 (Krizhevsky & Hinton, 2009), Cifar100 (Krizhevsky & Hinton, 2009), Tiny-ImageNet (Chrabaszcz et al., 2017), and Flower102 (Nilsback & Zisserman, 2008). All datasets are taken as 75% as training set and 25% as test set. *Data Heterogeneity:* For each dataset, we apply two main types of Non-IID setting (Zhang et al., 2024b): 1) Dirichlet distribution, we set $\alpha = \{0.1, 0.5\}$, it's worth noting that $\alpha$ is the concentration parameter and smaller $\alpha$ corresponds to stronger data heterogeneity; 2) Pathological Setting, each client contains a fixed number of sample categories, but each client has a different sample category. *Model Heterogeneity:* We adopt the heterogeneous model setup from (Zhang et al., 2021), including AlexNet (Krizhevsky et al., 2017), ShuffleNetV2 (Ma et al., 2018), ResNet18 (He et al., 2016) and GoogleNet (Szegedy et al., 2015), which are randomly and evenly assigned to clients. For the DeL experimental setup, we use only 2-layer CNNs and apply two-stage elastic distillation for homogeneous models (Li et al., 2022a).

**Baselines and Implementation**

*FL setting:* We select eight baselines, excluding Fed-Proto (Tan et al., 2022), FedTGP (Zhang et al., 2024a), RethinkFL (Huang et al., 2023), FML (Shen et al., 2020), FedKD (Wu et al., 2022), FedType (Wang et al., 2025), FedDistill (Jeong et al., 2023), HYDRA-FL (Khan et al., 2024) and FCCL (Huang et al., 2022). *DeL setting:* We add DPSGD (Lian et al., 2017), CGA (Esfandiari et al., 2021), Coll-FL (Zhu et al., 2023), L2C (Li et al., 2022a) and DeSA (Huang et al., 2025a) in a DL experiment. We evaluate the superiority of the algorithm by evaluating the average of all client accuracy and standard deviations throughout the training process. All experiments are implemented in Pytorch (i.e. Intel(R) Xeon(R) Platinum 8176 CPU @ 2.10GHz with NVIDIA GeForce RTX 3090 GPU).

**Security Settings**
To test the reliability of validation in a blockchain-based FL setting, we simulate malicious nodes through three methods: label reversal attacks, model noising to emulate malicious models (Qin et al., 2024), and logits attacks (Khan et al., 2024), evaluating robustness under these adversarial conditions.

## 5.2 Comparison with State-of-the-Art Methods

**Heterogeneous model.** We conduct experiments across four datasets and two data partitioning schemes, with results detailed in Tab.1($a.$). Our method consistently outperforms others by at least 2% across all experimental settings, with a 6.01% accuracy gain over FedTGP on Flower102 (Dirichlet 0.5). This highlights our temperature strategy's effectiveness in optimizing client gradient descent and enhancing knowledge transfer across heterogeneous models. Moreover, as shown in Fig.1a, our method achieves optimal

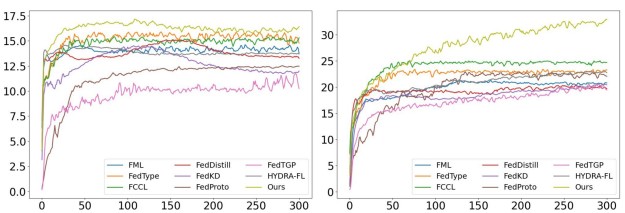
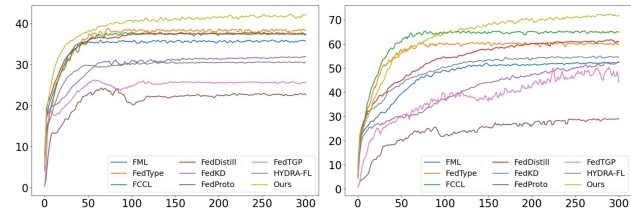

(a) Accuracy curves for Dir (0.1) distribution tested on Tiny-ImageNet (left), Flower102 (right).

(b) Accuracy curves for PAT distribution tested on Tiny-ImageNet (right), Flower102 (left).

*Figure 1.* Accuracy curves for PAT distribution tested on Tiny-ImageNet, Flower102.

*Table 3.* The testing accuracy (%) of heterogeneous model combination on Cifar-10. $\mathcal{A}$. for AlexNet, $\mathcal{R}$. for ResNet18, $\mathcal{G}$. for googleNet, $\mathcal{S}$. for ShuffleNetv2

| | $\mathcal{A}$. | $\mathcal{R}$. | $\mathcal{G}$. | $\mathcal{S}$. | $\mathcal{A}.+\mathcal{R}$. | $\mathcal{G}.+\mathcal{R}$. | $\mathcal{S}.+\mathcal{R}$. | $\mathcal{A}.+\mathcal{R}.+\mathcal{G}$. | $\mathcal{A}.+\mathcal{R}.+\mathcal{S}$. | $\mathcal{U}(All)$. |
|---|---|---|---|---|---|---|---|---|---|---|
| Accuracy | 89.28% | 89.04% | 89.38% | 86.69% | 88.94% | 89.16% | 87.84% | 88.64% | 87.49% | 88.79% |

*Table 4.* Ablation studies of our method on four datasets.

| Method | Cifar10 | Cifar100 | Flower102 | Tiny. |
|---|---|---|---|---|
| Single-level & Fixed $\tau$ | 66.85 | 27.52 | 26.97 | 13.20 |
| Multi-level & Fixed $\tau$ | 69.93 | 29.12 | 30.26 | 14.56 |
| Multi-level & Elastic $\tau$ | 70.82 | 29.45 | 32.05 | 15.14 |
| **ReT-FHD(Ours, FL)** | **72.76** | **30.77** | **33.16** | **17.24** |

results with fewer communication rounds, reaching near-peak performance on Tiny-ImageNet in under 80 rounds. In Fig.1b, while our method exhibits slower initial growth on Flower102, it ultimately surpasses other baselines in later rounds.

**Homogeneous model.** Tab.1(b) shows our method outperforms leading FL baselines across four ResNet-based datasets, achieving a 19.03% accuracy gain over HYDRA-FL on Flower102 under PAT's high-variability data. This highlights our framework's strength in handling feature ambiguity via elastic temperature scaling.

**DeL Experiments.** Our method employs blockchain for decentralized federated training, eliminating the need for a central server. We compare it with state-of-the-art decentralized methods in Tab.2. On CIFAR-10, our method and DeSA fall short of L2C due to its efficient convergence on simpler datasets. However, on the more complex CIFAR-100, our method achieves the highest accuracy, demonstrating its strength in extracting meaningful patterns. Additionally, blockchain integration enhances privacy protection in distributed communication.

### 5.3 In-depth Analysis

**Ablation studies.** We design a series of ablation experiments to verify the effects of Multi-level distillation, Elastic Temperature $\tau$ and Global Logits Scaling on model per-

formance. The experiment removes these key components sequentially and observes their impact on the final performance. Specifically, we start from the full model (Ours: Multi-level + Elastic Temperature $\tau$ + Global Logits Scaling) and sequentially remove Global Logits Scaling, Elastic Temperature $\tau$, and ultimately simplify it to a single-level fixed temperature (Single-level + Fixed $\tau$) of the simplest version. As shown in Tab.4 and Fig.2(left), the full model performs the best out of all the setups, achieving a performance of 72.75%. At the same time, it shows that Multi-level Distillation can initially improve the quality of distillation, while Elastic Temperature $\tau$ and Gloabl Logits Scaling can provide more knowledge in distributed learning compared with Fixed $\tau$.

**Evaluation of Distillation Layers Number $L$.** To investigate the impact of the number of distillation layers on model performance, we conduct experiment using the ResNet-18 architecture, varying the number of layers $L$ from 1 to 5. Fig.2(right) shows that increasing distillation layers from 1 to 4 enhances accuracy: CIFAR-10 boosts from 67.21% to 72.75% (+5.54), CIFAR-100 from 28.01% to 30.77% (+2.76), and Tiny-ImageNet from 14.21% to 17.24% (+3.03). These findings reflect diminishing returns with excessive layers, highlighting the need to match distillation layers to model complexity for optimal performance.

**Effect of different heterogeneous models on accuracy.** Tab. 3 shows the performance with heterogeneous model combinations on CIFAR-10. Multi-branch distillation achieves optimal accuracy (89.16%) using complementary architectures like GoogleNet ($\mathcal{G}$) + ResNet18 ($\mathcal{R}$). In contrast, combining all models ($\mathcal{U}$) degrades performance (88.79%), underperforming standalone $\mathcal{G}$. Results emphasize strategic model complementarity over quantity for effective FL.

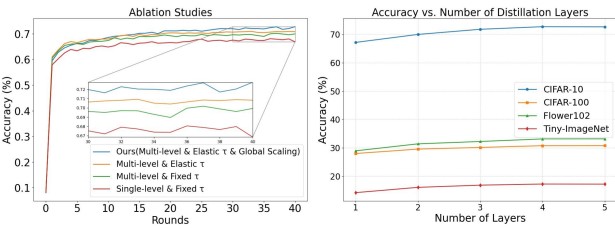

*Figure 2.* Ablation study (left) and Evaluation of Distillation Layers Number L (right) on Cifar-10.

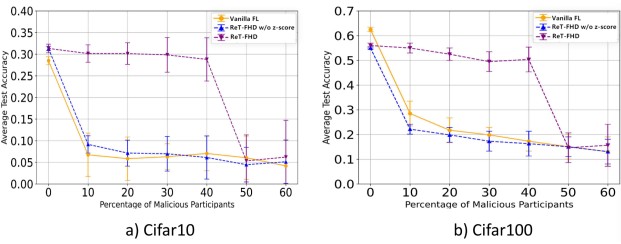

*Figure 3.* Security testing in a blockchain framework

**Analysis of Complexity and Communication Volume.** As shown in Tab.5, we analyze the communication upload and download data volumes, as well as the training complexity of all baseline methods. The experiments are conducted on the CIFAR-10 dataset, encompassing four different modalities in a heterogeneous setting (with the proxy model being ResNet-18). Overall, methods based on proxy models generate the highest communication volume due to the need to transmit model parameters. In contrast, our logits distillation method results in the least communication overhead. Regarding computational complexity, proxy model methods incur a significant computational burden during the local knowledge transfer between the teacher and student. In comparison, the prototype-based and logits distillation setups exhibit relatively lower computational demands during local training. In summary, our method outperforms other approaches in terms of both communication volume and computational complexity.

### 5.4 Security and Robustness Performance

To establish a reliable global logits benchmark, we configure 20 clients, each responsible for independent training on the CIFAR-10 dataset. We aggregate all logits of the same class using a weighted average to obtain the Z-scores for each category. These Z-scores are then visualized, with diagonal lines indicating predictions for the correct classes, demonstrating that the Z-scores for correct predictions are generally the highest.

**Malicious Model.** The experimental results in Tab.6 analyze the behavior of a malicious node in a simulated distributed network. By injecting noise into a ResNet model's

*Table 5.* Comparison of the mean communication cost and amount of computation per round for each approach. "MB" are short for megabyte.

| Methods | Comm.(MB) | | Computation(Gflops) |
|---|---|---|---|
| | Up. | Down. | |
| FedProto | 0.00256 | 0.00256 | 11340 |
| FedTGP | 0.00256 | 0.00256 | 11680 |
| FML | 10.45 | 10.45 | 23310 |
| FedKD | 10.45 | 10.45 | 4620 |
| FedType | 10.45 | 10.45 | 5160 |
| FedDistill | 0.003 | 0.003 | 11760 |
| DeSA | 0.6 | 0.6 | 24680 |
| **ReT-FHD(Ours)** | 0.0003 | 0.0004 | 3780 |

*Table 6.* Security Evaluation of Logits and Z-scores under Malicious Node and Label Reversal Attacks on CIFAR-10. This table presents the logits and their corresponding Z-scores under two adversarial attack scenarios: (1) malicious nodes injecting anomalous data for the "Airplane" and "Bird" classes, and (2) label reversal attacks flipping the labels between "Deer" and "Dog" classes.

| | | Malicious model | | | | Label reversal attack | | |
|---|---|---|---|---|---|---|---|---|
| | | Air | Auto | Bird | | Cat | Deer | Dog |
| $\mathcal{L}/\mathcal{C}$ | Airplane | **681.74** | 248.60 | 1805.4 | Deer | 1.7862 | 0.9026 | 3.4350 |
| | Bird | 0.0477 | -1.0924 | **0.0016** | Dog | 1.7653 | 3.3860 | 0.9474 |
| $Z_m$ | Airplane | **0.7710** | 0.1043 | 2.5008 | Deer | 1.1481 | 0.5623 | 2.2411 |
| | Bird | 0.5957 | -0.1628 | **2.6090** | Dog | 1.1441 | 2.2295 | 0.5963 |
| | $\Delta Z_m^g$ | 4.8235 | 3.6608 | 22.485 | $\Delta Z_m^g$ | 2.8462 | 10.203 | 18.7153 |
| $Z_{\mathcal{V}}$ | Airplane | 0.7203 | -0.6765 | -0.6595 | Deer | 0.7833 | 1.6304 | 1.0069 |
| | Bird | 0.3182 | -0.7181 | -0.7223 | Dog | 0.7682 | 0.8572 | 1.8016 |
| | $\Delta Z_{\mathcal{V}}^m$ | 2.3946 | 4.0729 | 61.4704 | $\Delta Z_{\mathcal{V}}^m$ | 2.8765 | 7.6342 | 7.5652 |
| | $\Delta Z_{\mathcal{V}}^g$ | 4.4746 | 3.0139 | 5.6661 | $\Delta Z_{\mathcal{V}}^g$ | 2.0812 | 2.9867 | 5.2146 |

*Table 7.* Testing accuracy(%) under attack and no attack.

| Methods | no attack | | | attack | | |
|---|---|---|---|---|---|---|
| | Flow. | Cif-100 | Cif-10 | Flow. | Cif-100 | Cif-10 |
| FedDistill | 20.47 | 21.04 | 63.51 | 10.31 | 9.28 | 20.75 |
| FCCL | 25.20 | 22.72 | 64.29 | 12.88 | 9.06 | 31.18 |
| DeSA | 30.18 | 26.84 | 65.68 | 12.89 | 9.82 | 29.14 |
| HYDRA-FL | 22.27 | 20.82 | 61.74 | 9.65 | 11.04 | 36.94 |
| **ReT-FHD(Ours, FL)** | 33.16 | 30.77 | 72.76 | 20.11 | 19.28 | 38.75 |

weights, we examine its impact on logits and Z-scores for randomly selected clients. For the first three CIFAR-10 categories, the malicious node's logits show irregular values, such as abnormally high scores for "Airplane." Similarly, its Z-scores $Z_m$ exhibit unusual patterns. To quantify deviations, we use $\Delta Z_m^g$ and $\Delta Z_{\mathcal{V}}^m$ (calculated via Eq.7), measuring differences between global, verification, and malicious Z-scores. While verification and global Z-scores remain normal, the malicious node's scores are significantly elevated, with $\Delta Z_m^g$ and $\Delta Z_{\mathcal{V}}^m$ confirming its aberrant behavior.

**Label Reversal Attack.** In Tab.6, we simulate label inversion by flipping "Deer" and "Dog" labels on a CIFAR-10 client. The malicious node shows skewed logits, with abnormally high confidence for "Dog" on deer samples, while the verification node's Z-score deviation from the global average remains minimal. This divergence in logit distributions and Z-score alignment confirms our system's efficacy in detecting label-flipping attacks.

**Logits Attack.** Clients under anomalous attacks produce logits conflicting with the optimization goal, disrupting federated learning distillation. Tab.7 analyzes test accuracy under client anomaly attacks on state-of-the-art logits distillation methods across three datasets. Our method outperforms others, with its multi-level distillation effectively mitigating logits attacks, as supported by prior research (Huang et al., 2025a). While HYDRA-FL also uses Multi-level distillation, it focuses on model homogeneity and underperforms in heterogeneous settings. This confirms that our dynamic temperature strategies enhance robustness and security in federated distillation.

**Accuracy and Poisoning Tolerance.** Model accuracy decrease as the percentage of malicious participants increased, with CIFAR-100 showing a sharper decline due to its complexity. In Fig. 3.a (CIFAR-10), accuracy dropped from 0.35 to below 0.1 with 60% malicious participants, while Fig. 3.b (CIFAR-100) shows a decline from 0.65 to under 0.2. These results demonstrate that server trust alone cannot mitigate adversarial impacts in federated learning, necessitating robust detection mechanisms and adaptive defenses to preserve performance under attack.

## 6 Conclusion

In this paper, we propose a novel dynamic temperature control strategy for federated distillation (ReT-FHD) and implement it in all FL, Del and blockchain-driven FL frameworks. Through comprehensive experiments, our approach demonstrates substantial improvements across three critical dimensions: model performance, communication efficiency, and security robustness. While our method addresses data and model heterogeneity, the current framework does not evaluate compatibility with attention-based architectures such as Vision Transformers (ViT) (Dosovitskiy et al., 2021). Future efforts will extend validation to Transformer frameworks given their prevalence in visual tasks.

## Acknowledgements

This work was supported by NSFC (No.62206200, 62206137, 62036012, 62376196 and U23A20387).

## Impact Statement

This paper presents work whose goal is to advance the field of Machine Learning. There are many potential societal consequences of our work, none which we feel must be specifically highlighted here.

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

# A Appendix

## A.1 The proof of Theorem 4.1 in Sec.4

For a discrete random variable $X$ with probability distribution $P(x)$, the information entropy is defined as:

$$H(X) = -\sum_{x \in X} P(x) \log P(x) \tag{15}$$

where P(x) is obtained from the softmax function. The softmax function is defined as follows:

$$\text{softmax}(p_i) = \frac{e^{z_i/T}}{\sum_{j=1}^{n} e^{z_j/T}} \tag{16}$$

Firstly, we can calculate the derivative of the probability distribution $p_i$ with respect to the temperature $T$:

$$\frac{\partial p_i}{\partial T} = \frac{\partial}{\partial T} \left( \frac{\exp(z_i/T)}{\sum_{j=1}^{n} \exp(z_j/T)} \right) \tag{17}$$

By calculation, it is possible to obtain:

$$\frac{\partial p_i}{\partial T} = \left[ \sum_{j=1}^{n} \exp\left(\frac{z_j}{T}\right) \exp\left(\frac{z_i}{T}\right) \left(-\frac{z_i}{T^2}\right) - \exp\left(\frac{z_i}{T}\right) \right.$$
$$\left. \cdot \sum_{j=1}^{n} \exp\left(\frac{z_j}{T}\right) \left(-\frac{z_j}{T^2}\right) \right] / \left( \sum_{j=1}^{n} \exp\left(\frac{z_j}{T}\right) \right)^2 \tag{18}$$

Simplifying and organizing gives:

$$\frac{\partial p_i}{\partial T} = p_i \left( \frac{\sum_{j=1}^{n} p_j z_j - z_i}{T} \right) \tag{19}$$

Secondly, information entropy is a fundamental concept used to measure the uncertainty of a random variable. For a discrete random variable $X$ with probability distribution $P(x)$, the information entropy is defined as:

$$H(X) = -\sum_{x \in X} P(x) \log P(x) \tag{20}$$

The information entropy is derived with respect to temperature as follows:

$$\frac{\partial H}{\partial T} = -\sum_{i=1}^{n} \left( \frac{\partial p_i}{\partial T} \log p_i + p_i \frac{\partial}{\partial T} (\log p_i) \right) \tag{21}$$

Simplifying and organizing gives:

$$\frac{\partial H}{\partial T} = -\sum_{i=1}^{n} \left( \frac{\partial p_i}{\partial T} \log p_i + \frac{\partial p_i}{\partial T} \right)$$
$$= -\sum_{i=1}^{n} \frac{\partial p_i}{\partial T} (\log p_i + 1) \tag{22}$$

By substituting the expression for $\frac{\partial p_i}{\partial T}$, we get:

$$\frac{\partial H}{\partial T} = -\sum_{i=1}^{n} p_i \left( \frac{\sum_{j=1}^{n} p_j z_j - z_i}{T} \right) (\log p_i + 1) \tag{23}$$

Based on the above derivation, the equation for the derivative of the entropy $H$ with respect to the temperature $T$ is known. Next, it is necessary to replace the conventional temperature $T$ with our predefined elastic temperature $\tau$ in Eq. (6). To accurately determine the relationship between entropy and the elastic temperature $\tau$, it is necessary to understand how $\tau$ varies with $\Delta Z$, namely $\frac{\partial \tau}{\partial \Delta Z}$.

By the definition of $G$, we can derive:

$$
\begin{aligned}
\frac{\partial \tau}{\partial \Delta Z} &= \frac{\partial}{\partial \Delta Z}\left(\xi + \frac{\gamma \log(1 + \Delta Z)}{\log(1 + \Delta Z_{\max})}\right) \\
&= \frac{\gamma}{\log(1 + \Delta Z_{\max})} \cdot \frac{1}{1 + \Delta Z}
\end{aligned}
\tag{24}
$$

Ultimately, the derivative of the entropy $H$ with respect to $\tau$ can be expressed as:

$$
\begin{aligned}
\frac{\partial H}{\partial \Delta Z} &= \frac{\partial H}{\partial \tau} \cdot \frac{\partial \tau}{\partial \Delta Z} \\
&= \left(-\sum_{i=1}^{n} p_i \left(\frac{\sum_{j=1}^{n} p_j z_j - z_i}{\tau}\right)(\log p_i + 1)\right) \\
&\quad \cdot \left(\frac{\gamma}{\log(1 + \Delta Z_{\max})} \cdot \frac{1}{1 + \Delta Z}\right)
\end{aligned}
\tag{25}
$$

The condition $\frac{\partial H}{\partial \tau} > 0$ implies that an increase in the elastic temperature $\tau$ leads to a decrease in the system's entropy. This reduction in entropy is reflected as an increase in information, known as information gain. Consequently, appropriately increasing the elastic temperature can enhance information gain, allowing the model to learn more effectively from the training data.

Then the Theorem 4.1 is proved.

## A.2 The proof of Corollary 4.2 in Sec. 3

For a model $f$ with random distribution $I$ in the initial state, for a deep neural network (DNN) model f with random distribution I in the initial state, the model has no preference for any sample. Thus for the model's predictive distribution $f(x; \theta_I) = \{p_1^I, p_2^I, ..., p_c^I\}$, the following relationship exists:

$$
\mathbb{E}(p^I(x)) = \mathbb{E}(p_1^I(x)) = \mathbb{E}(p_2^I(x)) = ... = \mathbb{E}(p_c^I(x))
\tag{9}
$$

**Assumption A.1.** We assume that the model is optimised under the guidance of a soft label distribution with smoothing degree $\lambda_1$ is satisfied:

$$
\begin{aligned}
&p_1^{\lambda_1}(x_1) = p_2^{\lambda_1}(x_2) = \cdots = p_c^{\lambda_1}(x_c) > \frac{1}{C} > \\
&p_1^{\lambda_1}(x_{c \neq 1}) = p_2^{\lambda_1}(x_{c \neq 2}) = \cdots = p_c^{\lambda_1}(x_{c \neq c})
\end{aligned}
\tag{10}
$$

For simplicity, we only consider a dataset that includes two categories, for which the categories are defined as follows.

**Definition A.2.** Let the cross-entropy loss of a sample be denoted as $\mathcal{L}_{\mathrm{CE}}$, and the average cross-entropy loss over all samples be $\bar{\mathcal{L}}_{\mathrm{CE}}$. The class ID is defined as:

$$
\text{class ID} = \begin{cases} 1, & \text{if } \mathcal{L}_{\mathrm{CE}} > \bar{\mathcal{L}}_{\mathrm{CE}}, \\ 2, & \text{otherwise.} \end{cases}
$$

Referring to the study of (Zhao et al., 2024), we introduce the knowledge scale to quantify the level of knowledge of the predictive distribution, which can measure the scale of knowledge learnt from the probability distribution by the model with randomly initialised weights:

$$
K^T = KL(I(x), T(x)) = \log C - H(P^T)
\tag{11}
$$

where $KL(\cdot)$ stands for Kullback–Leibler divergence, which calculates the cost of information required to travel from one distribution to another, $C$ is the total number of categories and is a constant for the same dataset, and $H(\cdot)$ is the information entropy.

First we calculate the derivative of the knowledge scale with respect to logits $z_c(x)$:

$$
\frac{\partial KL(f(x;\theta_I), P_{\lambda_1})}{\partial z_c(x)}
$$
$$
= \sum_{i=1}^{C} \frac{\partial p_i}{\partial z_c(x)} \cdot \frac{\partial KL(f(x;\theta_I), P_{\lambda_1})}{\partial p_i}
$$
$$
= \sum_{i=1}^{C} p_i^{\lambda_1}(x) p_c^I(x) - p_c^{\lambda_1}(x)
$$
$$
= p_c^I(x) - p_c^{\lambda_1}(x) \tag{12}
$$

For simplicity, we will subsequently develop the proof using $C = 2$ as an example. So we can easily obtain:

$$
\mathbb{E}\left( \frac{\partial KL(f(x_1;\theta_I), P_{\lambda_1})}{\partial z_1(x_1)} \right) = \mathbb{E}\left( \frac{\partial KL(f(x_2;\theta_I), P_{\lambda_1})}{\partial z_2(x_2)} \right)
$$
$$
= \mathbb{E}\left( p_c^I(x_c) - p_1^{\lambda_1}(x_1) \right) = \mathbb{E}\left( p_c^I(x_c) - p_2^{\lambda_1}(x_2) \right) \tag{13}
$$

$$
\mathbb{E}\left( \frac{\partial KL(f(x_1;\theta_I), P_{\lambda_1})}{\partial z_2(x_1)} \right) = \mathbb{E}\left( \frac{\partial KL(f(x_2;\theta_I), P_{\lambda_1})}{\partial z_1(x_2)} \right)
$$
$$
= \mathbb{E}\left( p_c^I(x_c) - p_2^{\lambda_1}(x_1) \right) = \mathbb{E}\left( p_c^I(x_c) - p_1^{\lambda_1}(x_2) \right) \tag{14}
$$

Combined with the previous assumptions, we can obtain that the partial derivatives of knowledge scale changes are symmetric for all categories, which suggests that the overall change in knowledge scale has a balanced impact on different categories. Therefore, our elastic temperature focus on the adjustment of knowledge scale at different stages of the model and treat different categories equally, which helps us to bridge the structural differences between heterogeneous models of different clients.

Then the Corollary 4.2 is proved.

### A.3 The proof of Theorem 4.3 in Sec.4

We explore the relationship between changes in the category gradient and temperature, using category 1 as an example.The gradient for category 1 is:

$$
\nabla_{f \sim \lambda_1}^1 = \mathbb{E}\left[ \left| \frac{\partial z_1}{\partial \theta} \cdot \frac{\partial KL\left(f_s(x_1;\theta_I), P_{\lambda_1}\right)}{\partial z_1} \right| + \left| \frac{\partial z_2}{\partial \theta} \cdot \frac{\partial KL\left(f_s(x_1;\theta_I), P_{\lambda_1}\right)}{\partial z_2} \right| \right] \tag{2}
$$

We increase the temperature of category 1 to $\tilde{\tau}_1 + \Delta\tilde{\tau}_1$ to obtain the global logits distribution $P_{\lambda_2}$.We can then obtain:

$$
\nabla_{f \sim \lambda_2}^1 - \nabla_{f \sim \lambda_1}^1 =
$$
$$
\mathbb{E}\left[ \left| \frac{\partial z_1}{\partial \theta} \cdot \frac{\partial KL\left(f_s(x_1;\theta_I), P_{\lambda_2}\right)}{\partial z_1} \right| + \left| \frac{\partial z_2}{\partial \theta} \cdot \frac{\partial KL\left(f_s(x_1;\theta_I), P_{\lambda_2}\right)}{\partial z_2} \right| \right]
$$
$$
- \mathbb{E}\left[ \left| \frac{\partial z_1}{\partial \theta} \cdot \frac{\partial KL\left(f_s(x_1;\theta_I), P_{\lambda_1}\right)}{\partial z_1} \right| + \left| \frac{\partial z_2}{\partial \theta} \cdot \frac{\partial KL\left(f_s(x_1;\theta_I), P_{\lambda_1}\right)}{\partial z_2} \right| \right] \tag{3}
$$

where $\frac{\partial z}{\partial \theta}$ can be regarded as the impact of the class itself on model weight optimization, in a practical sense, this part reflects the inner relationship between DNN and sample, more specifically, it can reflect the difficulty of the sample itself for

the model and not relate to the optimization object. Therefore, the above equation can be simplified as:

$$\nabla^1_{f\sim\lambda_2} - \nabla^1_{f\sim\lambda_1} =$$
$$\mathbb{E}\left[\frac{\partial z_1}{\partial\theta}\cdot\left(p_1^{\lambda_1}(x_1) - p_1^{\lambda_2}(x_1)\right) + \frac{\partial z_2}{\partial\theta}\cdot\left(p_2^{\lambda_1}(x_1) - p_2^{\lambda_2}(x_1)\right)\right] \tag{4}$$

Based on the effect of temperature on the probability distribution: $p_1^{\lambda_1}(x_1) - p_1^{\lambda_2}(x_1) > 0, p_2^{\lambda_1}(x_1) - p_2^{\lambda_2}(x_1) < 0$. At the same time, since category 1 is the main category, its gradient changes dominate. Therefore we can get:

$$\nabla^1_{f\sim\lambda_2} - \nabla^1_{f\sim\lambda_1} > 0 \tag{5}$$

The same can be obtained when we reduce $\Delta\tilde{\tau}_1$ for the category temperature to get the global logits distribution $P_{\lambda_3}$:

$$\nabla^1_{f\sim\lambda_3} - \nabla^1_{f\sim\lambda_1} < 0 \tag{6}$$

Then the Theorem 4.3 is proved.

### A.4 Blockchain-based FL Details

In the blockchain network, the role of the $v$ node is exclusively dedicated to the voting processes. This node evaluates transactions based on the established Equation $x$, which serves as a decision-making criterion. Depending on the results derived from applying this equation, the verification node can make its decisions, opting either for $P$ (Positive) or $N$ (Negative). This dichotomy allows the node to actively participate in the consensus mechanism by approving or rejecting transactions, thereby ensuring the reliability and security of the blockchain.

Subsequently, the node $v$ will take on the responsibility of broadcasting all the votes it has received for the node $w$ to all $m$ nodes within the network. This dissemination ensures that every $m$ node in our simulated network receives the set of values $z$ belonging to the set $\mathcal{Z}$ for each $w$. This comprehensive distribution of vote information across the network not only enhances transparency but also enables each node to have a holistic view of the network's consensus on each $w$, facilitating more informed decision-making and maintaining the integrity of the overall system.

Each node $m$ in the network meticulously processes all data extracted from the transactions sent by node $v$. Upon processing, each $m$ node synthesizes this information into a series of candidate blocks. A typical candidate block, denoted as $block_j^m$, encapsulates not only the global logits $z_i$ but also includes the result $R$ and data pertaining to the blacklist $\mathcal{B}$. This blacklist is dynamically generated based on the collective voting outcomes, reflecting the network's consensus on disreputable nodes or transactions.

Both types of nodes, $v$ from set $\mathcal{V}$ and $m$ from set $\mathcal{M}$, receive rewards in alignment with their contributions to the network's security and integrity. The reward system is designed to incentivize activities that uphold the blockchain's operational standards and trustworthiness. Furthermore, the blockchain protocol is engineered to select the candidate block from the most affluent $m$ nodes—the ones with the highest accrued rewards—as the preferred choice for linkage to the blockchain. This selection criterion ensures that the most reliable and trustworthy nodes, as determined by their wealth accumulation through valid and honest contributions, have their candidate blocks prioritized in the blockchain extension process.

To effectively simulate a real blockchain-based Federated Learning (FL) environment, our methodology includes the dissemination of pertinent information through the network. This is achieved by encapsulating the information within a transaction, which is then securely signed using the private key of the designated role. This approach ensures the integrity and authenticity of the data being broadcast across the network.

### A.5 Additional Experimental Results

**Effects of different E and K.** In our experiments, we specifically assess the performance of our method across different setups, involving 50 and 100 clients. The results, detailed in Table 8, indicate that while there is a slight decline in accuracy as the number of clients increases, our method demonstrates relatively stable outcomes over multiple training cycles. For instance, with 50 clients, our method achieve an accuracy of 42.46% at $E = 20$, and with 100 clients, it maintains a respectable 37.57%, despite the increased complexity and potential for data discrepancies among a larger number of clients. In contrast to the general trend observed with other models, our proposed High-Dimensional Federated Learning (HDFL) approach managed to maintain performance across an extended number of client training cycles. This resilience

*Table 8.* The test accuracy (%) on Cifar100 in the practical setting using the heterogeneous model group with a different number of client training epochs (E) and different number of clients(K).

|  | $E = 5$ | $E = 10$ | $E = 20$ | $K = 50$ | $K = 100$ |
|---|---|---|---|---|---|
| FML | $39.63 \pm 0.83$ | $39.81 \pm 0.82$ | $39.45 \pm 0.83$ | $36.65 \pm 1.62$ | $33.78 \pm 1.66$ |
| FedProto | $37.61 \pm 0.91$ | $37.34 \pm 1.01$ | $35.71 \pm 1.31$ | $33.24 \pm 2.14$ | $31.79 \pm 1.89$ |
| FedDistill | $40.20 \pm 0.98$ | $39.82 \pm 0.97$ | $39.89 \pm 1.01$ | $37.77 \pm 1.60$ | $35.36 \pm 1.66$ |
| FedKD | $37.01 \pm 1.11$ | $36.96 \pm 1.12$ | $37.77 \pm 1.07$ | $32.64 \pm 1.85$ | $32.34 \pm 1.64$ |
| FedTGP | $42.87 \pm 1.67$ | $41.69 \pm 1.84$ | $40.94 \pm 1.80$ | $39.84 \pm 2.03$ | $36.75 \pm 1.84$ |
| **Ours** | $\mathbf{47.98 \pm 0.62}$ | $\mathbf{48.38 \pm 0.64}$ | $\mathbf{48.51 \pm 0.63}$ | $\mathbf{42.46 \pm 1.62}$ | $\mathbf{37.57 \pm 1.86}$ |

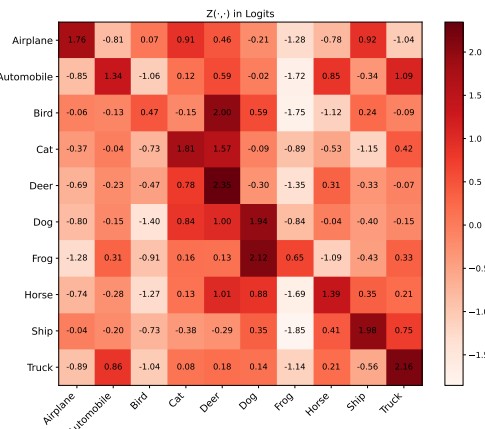

*Figure 4.* Global Logits Z-Score Visualization for Cifar-10 with 20 Clients

can be attributed to our model's ability to effectively manage and synchronize heterogeneous data and learning strategies across diverse client environments. The capability to sustain high accuracy levels under varying client counts and training conditions highlights the robustness of our approach, making it particularly suitable for practical applications where client variability and training dynamics are challenging factors.

**Global Logits Z-Score Visualization.** Fig.4 visualizes the Z-Score of the global logits under the Cifar-10 dataset. It can be found that the data distribution of logits is more concentrated through the z-score processing.

**Supplement of Tab.6.** Tab.9 and Tab.10 extends the results of Tab.7 by providing a comprehensive display of the complete logits and Z-scores, offering deeper insights into the behavior of the models under the evaluated scenarios. These supplementary results aim to enhance the understanding and robustness of the conclusions drawn from the main experiments.

**Complete Experimental Data on Malicious Model.** Tab.9 shows that the complete logits and Z-scores obtained by injecting noise to the weights of the malicious node's category airplane and category bird. The comprehensive analysis of logits and Z-scores across all categories in the supplementary data reveals distinct behavioral patterns between malicious and normal nodes. Specifically, the malicious node exhibits significant anomalies in the "Airplane" and "Bird" categories, with logits and Z-scores deviating markedly from the normal range (e.g., specific values in Z-scores for category airplane corresponding to category airplane and category bird: [0.7710, 2.5008] → [0.7203, -6595]). In contrast, other categories such as "Car" and "Trucks" remain stable, with minimal deviations (e.g., [-0.6385, -0.8651] vs. [-0.7638, -0.7107], $\Delta$ = [0.1253, -0.1544]). The differential measures $\Delta Z_m^g$ and $\Delta Z_{\mathcal{V}}^m$ effectively quantify these deviations, with values for anomalous categories (e.g., Bird: [22.485, 61.4704]) significantly exceeding those of normal categories. These findings underscore the targeted nature of the attack and the robustness of the proposed detection metrics in identifying malicious activities.

**Complete Experimental Data on Label Reversal Attack.** In Tab. 10, we simulate a label inversion attack by flipping the labels of "Deer" and "Dog" on a selected CIFAR-10 client. The malicious node exhibits significant anomalies in its logits, particularly displaying abnormally high confidence for the "Dog" class when the true label is "Deer." For instance, the Z-Score for "Dog" on deer samples rise to $2.2411$, compared to the normal range of $[-0.6547, 1.1481]$. In contrast, the

*Table 9.* Supplement to Table 6.(Malicious model)

| | | Airplane | Automobile | Bird | Cat | Deer | Dog | Frog | Horse | Ship | Trucks |
|---|---|---|---|---|---|---|---|---|---|---|---|
| $\mathcal{L}/\mathcal{C}$ | Airplane | 681.74 | 248.60 | 1805.43 | -233.92 | -289.58 | -133.81 | 117.65 | 105.17 | -111.56 | -381.13 |
| | Bird | 0.0477 | -1.0924 | 0.00163 | -0.03615 | -0.6927 | -0.0313 | -0.78591 | 0.11322 | 0.17006 | -0.0338 |
| $Z_m$ | Airplane | 0.7710 | 0.1043 | 2.5008 | -0.6385 | -0.7242 | -0.4844 | -0.0973 | -0.1165 | -0.4501 | -0.8651 |
| | Bird | 0.5957 | -0.1628 | 2.6090 | -0.7720 | -0.2781 | -0.6905 | -0.2938 | -0.1418 | -0.1322 | -0.7333 |
| | $\Delta Z_m^g$ | 4.8235 | 3.6608 | 22.485 | 7.0336 | 14.503 | 5.1918 | 8.1249 | 4.8195 | 5.7027 | 3.1847 |
| $Z_\mathcal{V}$ | Airplane | 0.7203 | -0.6765 | -0.6595 | -0.7638 | 0.4510 | 1.0750 | -0.6278 | -0.8247 | 2.0166 | -0.7107 |
| | Bird | 0.3182 | -0.7181 | -0.7223 | -0.5224 | 0.4126 | 1.6685 | -0.6531 | -0.8371 | 1.7418 | -0.6880 |
| | $\Delta Z_\mathcal{V}^m$ | 2.3946 | 4.0729 | 61.4704 | 2.4421 | 5.5224 | 17.1949 | 3.2096 | 4.1907 | 20.675 | 2.2256 |
| | $\Delta Z_\mathcal{V}^g$ | 4.4746 | 3.0139 | 5.6661 | 7.3089 | 6.3297 | 6.99128 | 5.1713 | 2.40193 | 8.0513 | 3.3053 |

*Table 10.* Supplement to Table 6.(Label reversal attack)

| | | Airplane | Automobile | Bird | Cat | Deer | Dog | Frog | Horse | Ship | Trucks |
|---|---|---|---|---|---|---|---|---|---|---|---|
| $\mathcal{L}/\mathcal{C}$ | Deer | -0.8718 | -0.3399 | -0.8518 | 1.7862 | 0.9026 | 3.4350 | -0.9332 | -0.9158 | -0.7916 | -0.8755 |
| | Dog | -0.9006 | -0.3496 | -0.8857 | 1.7653 | 3.3860 | 0.9474 | -0.9386 | -0.9114 | -0.7518 | -0.7910 |
| $Z_m$ | Deer | -0.6140 | -0.2614 | -0.6008 | 1.1481 | 0.5623 | 2.2411 | -0.6547 | -0.6432 | -0.5608 | -0.6165 |
| | Dog | -0.6413 | -0.2723 | -0.6313 | 1.1441 | 2.2295 | 0.5963 | -0.6667 | -0.6485 | -0.5416 | -0.5679 |
| | $\Delta Z_m^g$ | 2.2690 | 2.1762 | 3.3980 | 2.8462 | 10.203 | 18.7151 | 3.2746 | 4.6532 | 2.4289 | 3.3192 |
| $Z_\mathcal{V}$ | Deer | 0.3912 | -0.7767 | -0.6786 | 0.7833 | 1.6304 | 1.0069 | -0.6776 | -0.8615 | 0.3785 | -0.6293 |
| | Dog | 0.3617 | -0.7883 | -0.6498 | 0.7682 | 0.8572 | 1.8016 | -0.7156 | -0.8255 | 0.3113 | -0.5843 |
| | $\Delta Z_\mathcal{V}^m$ | 5.6487 | 4.0295 | 2.4296 | 2.8765 | 7.6342 | 7.5652 | 2.4054 | 2.9043 | 5.02901 | 2.3278 |
| | $\Delta Z_\mathcal{V}^g$ | 6.4078 | 4.3835 | 3.3734 | 2.0812 | 2.9867 | 5.2146 | 2.7634 | 6.7946 | 4.140 | 3.8648 |

verification node's Z-scores remain closely aligned with the global average, with smaller deviations (e.g., $\Delta Z_\mathcal{V}^g$ values within $[2.7634, 6.4078]$). This stark divergence in logit distributions and Z-score alignment between the malicious and verification nodes highlights the effectiveness of our system in detecting label-flipping attacks. Furthermore, the differential measures $\Delta Z_m^g$ and $\Delta Z_\mathcal{V}^m$ quantify the malicious node's deviations, with values for the flipped categories (e.g., $[7.6342, 7.5652]$) significantly exceeding those of normal categories. These results demonstrate that our approach not only identifies the presence of label-flipping attacks but also provides robust metrics for quantifying the extent of the adversarial manipulation.

