# OpenReview forum: "Rethinking the Temperature for Federated Heterogeneous Distillation"
_ICML.cc/2025/Conference — ICML 2025 poster_

### Official Review · Reviewer_e9EW · 2025-03-12

**Overall Recommendation:** 5

**Summary:**

This paper highlights the suboptimal temperature calibration issue in existing federated distillation (FD) methods. To address this, the authors propose ReT-FHD, which introduces multi-level elastic temperature to dynamically adjust distillation intensities across different model layers and category-aware global temperature scaling for class-specific calibration. Additionally, they integrate a Z-score Guard blockchain verification mechanism to defend against label-flipping and poisoning attacks. Extensive evaluations across multiple benchmarks validate the effectiveness of ReT-FHD, demonstrating its superiority over existing FD approaches.

**Claims And Evidence:**

The security protection related content in this article suffers from an inadequately articulated rationale and demonstrates weak contextual integration with the remainder of the content, ultimately coming across as a superfluous addition that compromises the paper's structural coherence.

**Essential References Not Discussed:**

Some key competitors are not discussed in the paper.

For proto-based method: Rethinking federated learning with domain shift: A prototype view[C]//2023 IEEE/CVF Conference on Computer Vision and Pattern Recognition (CVPR). IEEE, 2023: 16312-16322.

For attack defense: Self-driven entropy aggregation for byzantine-robust heterogeneous federated learning[C]//Forty-first International Conference on Machine Learning. 2024.

**Experimental Designs Or Analyses:**

1. The authors show abundant experiments to demonstrate the effectiveness of the proposed methods on heterogeneous data & models.
2. The experimental section lacks a comparative analysis between the proposed method and established attack-defense FL frameworks, which significantly undermines the empirical validation of ReT-FHD's effectiveness.
2. While the authors present a comparative analysis of communication efficiency, the blockchain overhead inherent to ReT-FHD remains notably absent from their evaluation framework.

**Methods And Evaluation Criteria:**

1. The proposed multi-level elastic temperature is based on layer-level distillation, which looks incremental but interesting.
2. The category-aware global temperature scaling component is difficult to understand, primarily due to the lack of annotations in the equations and an unclear description of the training strategy.
3. Section 3.3 is quite confusing, as it presents a large amount of information with weak contextual integration.

**Other Comments Or Suggestions:**

None.

**Other Strengths And Weaknesses:**

1. The notations in equations are not clearly described.
2. The axis titles are missing in Fig. 1.

**Questions For Authors:**

1. The whole structure of the paper is massive. There is no clear connection between elastic temperature scaling and the security components, and the rationale for incorporating security remains unclear.
2. Lack of annotation in equations makes the paper hard to understand.

**Relation To Broader Scientific Literature:**

The core idea of this paper builds on previous findings that layer-level distillation enhances performance. The authors investigate the impact of temperature at each layer and introduce elastic temperature scaling to further refine distillation quality.

**Theoretical Claims:**

I am not an expert in theoretical analysis. From my point of view, the theoretical claims in the paper are solid.

---

> ### Author Rebuttal · Authors · 2025-03-30
>
> We thank reviewer e9EW for the constructive comments: "…dynamically adjust distillation intensities across different model layers...", "…validate the effectiveness of ReT-FHD...". To thoroughly address your concerns, we will answer the questions one by one:
>
> **Questions:**
>
> **Q1: Connection between elastic temperature scaling and the security components, and the rationale for incorporating security**
> >Our motivation is to **explore whether logits can be applied to complex heterogeneous FL scenarios**. To this end, we adopt Z-Score to normalize the knowledge distillation process and further introduce multi-level distillation, elastic temperature scaling, and secure validation mechanisms for optimizing cross-architecture knowledge distillation and enhancing model robustness, respectively. In addition, the verification mechanism of the model-based blockchain FL approach is difficult to apply to the logits sharing scenario. Therefore, we design a Z-Score-based verification mechanism, which is customized for the logits sharing FL method, which ensures efficient logits-based communication and effectively defends against malicious attacks.
>
> **Q2: Lack of annotation in equations**
> >- **Eq.3** describes the multistage elastic temperature mechanism, where each stage of the model is assigned an independent temperature adjustment. The temperature of each stage of the model is adjusted according to the logarithmic difference of the next stage. The temperature is bounded by $\xi$ to maintain stability and prevent degradation. The scaling factor $\gamma$ controls sensitivity. $\Delta Z_l$ represents the difference between current and next stage logits, guiding the temperature adjustment. $\Delta Z_{\max}$ normalizes the temperature across stages for consistency.
> >
> >- **Eq.4** describes the category-aware temperature scaling mechanism, which dynamically adjusts each category’s temperature based on its cumulative loss relative to the average loss. The weighting factor $\beta$ controls the sensitivity of temperature updates. $\mathcal{L}_{\mathrm{CE}}(f(\tilde{x}_c))$ represents the cumulative loss for category c, and the average loss across all categories serves as a global reference. The maximum absolute difference standardizes the temperature update across categories, ensuring consistency.
> >
> >- **Eq.5** defines the optimization objective of ReT-FHD. Here, $|\mathcal{D}|$ and $|\mathcal{D_i}|$ denote the total and local data sizes, respectively.
> $\mathcal{L}_{\mathrm{CE}}$ is the cross-entropy loss. The model applies KL divergence for knowledge distillation at each layer. $\sigma^t\left(\cdot\right)$ and $\sigma^s(\cdot)$ represent the softmax functions of the global and local logits, respectively. $\mathbf{z_t}$ denotes the global logits, $\bar{z_t}$ local logits' mean, and $\operatorname{Var}\left(\mathbf{z}_t\right)$ local logits' variance. $G(\Delta Z)$ and $\tilde{\tau}_c$ represent the temperature coefficients for model stages and categories, calculated using Eq.3 and 4, respectively. Unlike global logits, local logits are unaffected by category temperature adjustments.
>
> **Experimental Designs Or Analyses:**
> >Our validation mechanism, designed for logits-based scenarios, standardizes malicious logits detection and is mainly applicable to logits sharing, rather than model parameter-sharing FL frameworks. In security experiments, we demonstrate its effectiveness compared to other FL methods.
> On CIFAR-10, each validation (Eq.7) requires **~100-200 FLOPs**, while a ResNet18 forward pass takes **~1.8 GFLOPs**, making the blockchain overhead negligible. Moreover, since the blockchain mechanism only verifies logits without altering the FL communication process, ReT-FHD maintains the same communication efficiency as standard FL, as confirmed in Tab.5.
>
> **Essential References Not Discussed:**
> >Thanks for pointing out these references. Due to time constraints, we add additional experiments on **Cifar10** to more fully assess the effectiveness of ReT-FHD in dealing with model heterogeneity and safety. While **FPL[1]** mitigates domain shift, it may introduce information redundancy in our setting, whereas **SDEA[2]** is designed for homogeneous models and relies on high-quality public datasets.
> |Hetero. & No attack|RethinkFL|ReT-FHD|
> |---|---|---|
> |accuracy | 68.21% | 72.76% |
> |**Attack & Homo.**|**SDEA**|**ReT-FHD**|
> |accuracy|50.89%|51.75%|
>
> **Other Strengths And Weaknesses:**
> >If you have any doubts about specific equations, please refer to our response to Q2. As for Fig.1, the y-axis represents accuracy, and the x-axis indicates the number of communication rounds.
>
> **References:**
> >[1] Rethinking federated learning with domain shift: A prototype view
> >
> >[2] Self-driven entropy aggregation for byzantine-robust heterogeneous federated learning
>
> Thank you for dedicating your time and effort to reviewing our work. Please let me know if I have addressed your concerns and if not, we welcome any further questions you might have.

---

> > ### Comment · Reviewer_e9EW · 2025-04-03
> >
> > Thanks for your detailed answer. I took some more time to think about the points you raised:
> > 1. Based on your response, I understand the rationale for introducing security components. However, I still have some questions that need further clarification. What is the specific role of the Z-Score in the verification? How does it enhance the reliability?
> > 2. Function G is computed in Eq. 3 and applied in Eq. 5. However, I still have doubts about its specific role, particularly how it operates in Eq. 5.
> > The clarifications and additional details in the rebuttal addressed many of my concerns, and I would have been more inclined to raise my rating on the paper if the questions above had been explained.

---

> > > ### Author Response · Authors · 2025-04-04
> > >
> > > Dear reviewer, thank you for your response! Please find our responses below.
> > >
> > > **Q1: What is the specific role of the Z-Score in the verification mechanism? How does it enhance the reliability of verification?**
> > > >Z-Score normalization effectively mitigates the inconsistency in numerical scales caused by data and model heterogeneity, while preserving the original information of client logits. By eliminating absolute numerical differences, it ensures the comparability of logits across clients, providing a fairer and more robust foundation for the validation mechanism.
> > > >- After Z-Score normalization, the standard deviation of the logits becomes$1/\tau$, which normalizes the logits of each client to a zero-mean, Gaussian-like distribution while maintaining their relative relationships. This enhances the accuracy of the validation mechanism in detecting deviations in client logits and improves its sensitivity to malicious clients.
> > > >
> > > >- During the verification process, based on Z-Score normalization, we define the label set overlaps under heterogeneous data scenarios as:$|\mathcal{V} \cap \mathcal{H}|=\eta, |\mathcal{E} \cap \mathcal{H}|=\phi,
> > >  |\mathcal{V} \cap \mathcal{E}|=\psi$. These overlap metrics are used to weight the corresponding logits differences to further reduce the influence of heterogeneity during verification. Next, all clients' logits are scaled with a unified temperature $\tau$ to ensure consistency in distribution. The verification node then computes three types of differences: the difference between the verified node i and the verifier$\Delta Z_{\mathcal{E},\mathcal{V}}^i$, between the verified node and the global logits$\Delta Z_{\mathcal{E},g}^i$, and between the verifier and the global logits$\Delta Z_{\mathcal{V},g}^i$. Finally, a voting mechanism based on the weighted relationships among these differences is applied to enhance the reliability of the validation process.
> > >
> > > **Q2: How does function G work in Equation 5?**
> > > >The function G in Equation (3) computes the temperature coefficients at different stages of the model and is used in Equation (5) as part of the softmax function σ to adjust the logits' temperature:
> > > >- For global logits, the stage-wise temperature computed by G is multiplied by the class-specific temperature computed in Equation (4), jointly determining the final temperature coefficient. Accordingly, the general Z-Score formulation in Equation (2) is instantiated for global logits as:
> > > >>
> > > >>$$\mathcal{Z}\left(\boldsymbol{z_t} ; \tau_t\right)^{(c)} = \frac{\boldsymbol{z_t}^{(c)} - \bar{z_t}}{\operatorname{Var}\left({\boldsymbol{z_t}}\right)\cdot \tau_t}, \tau_t=G(\Delta Z)\cdot\tilde{\tau}_c$$
> > > >>
> > > >where $\boldsymbol{z_t}$ represents the global logits, $\tau_t$denotes the global logits' temperature, and other symbols follow the definitions in the previous explanation.
> > > >
> > > >- For local logits, since class-specific temperature does not apply to local logits, the temperature computed by G is used directly as the final temperature coefficient. This ensures flexibility and effectiveness in temperature adjustment during the knowledge distillation process. Thus, the general Z-Score formula in Equation (2) is instantiated for local logits as:
> > > >>
> > > >>$$\mathcal{Z}\left(\boldsymbol{z_s} ; \tau_s\right)^{(c)} = \frac{\boldsymbol{z_s}^{(c)} - \bar{z_s}}{\operatorname{Var}\left({\boldsymbol{z_s}}\right)\cdot \tau_s}, \tau_s=G(\Delta Z)$$
> > >
> > > We again thank the reviewers for the pertinent questions and helpful suggestions they gave us!

---

### Official Review · Reviewer_TEBy · 2025-03-12

**Overall Recommendation:** 3

**Summary:**

The paper introduces ReT-FHD, a framework for heterogeneous federated knowledge distillation that contributes three core ideas. First, it proposes Multi-level Elastic Temperature to adaptively regulate how much knowledge is distilled at each layer, enhancing cross-architecture consistency. Second, it implements Category-Aware Global Temperature Scaling, assigning class-specific temperatures to better handle non-IID data. Analysis shows this stabilizes convergence by harmonizing gradient variations. Finally, it integrates a Z-Score–based Guard to detect and deter malicious behaviors like label flipping. Experiments across benchmarks show significant accuracy improvements and communication efficiency.


**Update after rebuttal:**

 Thanks to the authors for their response. After reading it, I would like to keep my original score.

**Claims And Evidence:**

**Well-Supported Claims:**

**1. Adaptive Temperature Calibration Improves Accuracy.**

Claim: Multi-level elastic temperature and category-aware scaling enhance accuracy under model/data heterogeneity.

Evidence: Experiments on CIFAR-10/100, Tiny-ImageNet, and Flower102 show consistent accuracy gains. Ablation studies confirm the necessity of both components.

**2. Communication Efficiency.**

Claim: ReT-FHD reduces communication costs compared to proxy-based methods.

Evidence: Quantified communication overhead aligns with the claim.

**Problematic Claims:**

**1. Theoretical Guarantees (Theorem 4.1) on Temperature-Efficacy.**

Claim: Theorem 4.1 is presented as a theoretical foundation showing how temperature adjustments reduce entropy or gradient variance, thereby enhancing heterogeneous distillation.

Issue: The paper does not clarify under what specific assumptions (e.g., model randomness, uniform client participation, or particular distributions of logits) this theorem holds. It is also unclear whether the same conditions remain valid in the multi-level knowledge distillation or Category-Aware Global Temperature Scaling framework. From the proof in the appendix, it appears that the theorem’s validity depends on the specific algorithm proposed by the authors.

**2. Applicability of Lemma 4.2 in Realistic Scenarios.**

Claim: Lemma 4.2 assumes a random initialization where the model is indifferent to all classes, thereby implying balanced effects on each class under soft label distributions.

Issue: This assumption may be too strong for real-world settings, where data can be highly skewed and models might be partially pre-trained or not truly “randomly” initialized. As a result, the claimed uniform impact on different categories could fail to hold, and the lemma does not clarify how quickly or under what conditions the model transitions from this idealized initialization into a state where nontrivial data heterogeneity and model structure are present.

**Essential References Not Discussed:**

One area the paper could further contextualize is robust aggregators and Byzantine-resilient FL. Although the authors mention label-flipping and some blockchain-based checks, they do not engage extensively with the broader literature on strong Byzantine defense mechanisms. For example, seminal works like Krum, Bulyan, or other robust-mean estimators have laid out various theoretical guarantees. Since the authors’ proposal relies partly on logit-verification to detect anomalies, connecting their method to known robust aggregation strategies would help clarify whether and how ReT-FHD stands on par with or improves upon existing Byzantine-robust solutions.

Additionally, there is a growing body of work on dynamic or adaptive temperature in knowledge distillation (outside of FL contexts), such as “Adaptive Temperature for Distillation” or “Dynamic Soft Labels,” which the paper might draw upon to strengthen its theoretical rationale. Including those references would make the paper’s unique application of adaptive temperature to heterogeneous FL even clearer.

**Experimental Designs Or Analyses:**

**1. Heterogeneous Model Configuration:**

The paper states that four models (AlexNet, ShuffleNetV2, ResNet18, GoogleNet) are used in a heterogeneous setup. However, it does not detail exactly how these models are distributed among clients, nor whether some clients share the same architecture. Adding a clear baseline (e.g., no heterogeneity) for comparison would help highlight how well the proposed method handles real-world model diversity.

**2. Range of Dirichlet Parameters:**

The authors mainly present results for \alpha=0.5, with limited discussion of the more extreme alpha=0.1. It would be helpful to visualize or table out how performance changes as \alpha varies (e.g., 0.1, 0.3, 0.5) to confirm the method’s sensitivity to stronger non-IID data.

**3. Homogeneous vs. Heterogeneous in DeL:**

The decentralized setting (Table 2) uses only 2-layer CNNs, implying no model heterogeneity. It remains unclear how the method would perform under simultaneous data/model heterogeneity in a truly decentralized scenario. Presenting such an experiment would better demonstrate ReT-FHD’s robustness in the most challenging setups.

**4. Ablation Studies vs. Main Table Results:**

The ablation experiments in Table 4 report a final CIFAR-10 accuracy of 72.75%, which is close to the 72.76% in Table 1 under Dir(\alpha=0.5). However, it is unclear if those ablations are indeed run under the same Dirichlet parameters and the same heterogeneous model assignment. Moreover, the “Single-level & Fixed \tau” condition seems to combine multi-level knowledge distillation and Z-score distillation, but the paper does not indicate whether that configuration matches any baseline in Table 1. Clarifying these experimental conditions would help readers match ablation performance to main table comparisons.

**5. Data/Model Heterogeneity Parameters (Tables 5-7):**

The paper does not specify the exact Dirichlet settings or model architectures used for the experiments in Tables 5, 6, and 7 (e.g., how many clients used each architecture or the distribution of labels). Since these tables measure communication cost, malicious attack detection, and robustness, readers need a clear understanding of the data/model heterogeneity to interpret the results accurately; Table 5 omits a direct comparison with FCCL, another logit-based approach, leaving it uncertain how the proposed method stands against the most relevant baseline in terms of communication and computation. Also, the authors claim a notably lower cost than all other methods, but an explanation of why ReT-FHD, despite using additional temperature scaling steps, still results in reduced overhead would be beneficial.

**Methods And Evaluation Criteria:**

**1. Design Rationale for Key Equations:**

For Equation (3), the paper bases its temperature adaptation on the Z-score difference between the next-stage logits and teacher logits, yet does not explain why the difference involving the current-stage logits was not considered. If the goal is to measure the discrepancy between local and teacher knowledge, one might expect the current stage to be more directly relevant. For Equation (4), it applies a negative sign before \beta, effectively lowering the temperature for classes with higher loss. This design choice seems at odds with the intuition that “hard” classes (i.e., larger loss) should receive more guidance. A detailed rationale or additional experiments could clarify why reducing temperature for higher-loss classes aids learning.

**2. Selective Scaling of Only Global Logits:**

The paper confines category-aware temperature scaling to global logits, without mentioning the possibility of partially or fully scaling local logits. Since local distributions may be highly skewed, applying a similar scaling locally could potentially yield further improvements—or at least warrants discussion to justify why the authors limit scaling to the global side.

**3. Completeness of the Update Formula:**

Equation (6) is missing the intermediate update and lacks an explicit explanation of how the parameters are iteratively updated. Clarifying that step would better illustrate how multi-level distillation is integrated into the local optimization process.

**4. Motivation and Implementation Details:**

While the paper’s overarching goal—enhancing model performance via adaptive temperature—is clear, some formulaic details require further justification. Providing explicit motivations for these design choices (e.g., how each approach helps balance knowledge across architectures or handles non-IID distributions) would help readers understand why these decisions are suited to the specific problem of heterogeneous federated learning.

**Other Comments Or Suggestions:**

There are some typos:

1. In the paper, Algorithm 1 references color coding (e.g., “highlighted in orange/pink/purple”) to distinguish between FL, DeL, or blockchain steps, but there is no explicit legend or explanation for the orange portion. This can be confusing to readers trying to follow the workflow.

2. In line 262, the phrase “We assume that the model is optimised under the guidance of a soft label distribution with smoothing degree \lambda_1 is satisfied:” seems grammatically wrong.

3. Line 220 incorrectly refers to Corollary 1 instead of Corollary 4.1.

4. In Section 3.2, the text introduces $z_{t}^{(l)}$ (logits of the teacher) but never explicitly defines $z_{s}^{(l)}$.

**Other Strengths And Weaknesses:**

**Other Strengths**

1. Creative Combination of Techniques: The paper’s integration of elastic temperatures (multi-level and category-aware) and a Z-score–based blockchain security guard represents a novel synthesis of ideas from robust FL, KD, and malicious detection. Even if each concept (dynamic temperature, logit sharing, blockchain validation) has appeared in some form, combining them systematically to tackle heterogeneous FL highlights originality.

2. Applicability to Diverse FL Modes: By explicitly supporting centralized, decentralized, and blockchain-based topologies, the authors address a variety of practical deployment scenarios. This flexibility extends the potential reach of ReT-FHD to real-world federated systems that might not always rely on a single server or a purely decentralized approach.

**Other Weaknesses**

1. Limited Discussion of Other Cases: While the paper mentions Transformers or attention-based models as future work, it does not evaluate or discuss how easily ReT-FHD might extend to such architectures. This leaves open the question of whether the proposed solution is limited to some specific models or can generalize broadly.

2. Clarity Gaps: Although the presentation is overall coherent, a more detailed discussion of certain design choices (e.g., the exact model distributions for heterogeneous clients, the rationale behind measuring Z-score difference with the next stage rather than the current stage) would improve transparency and reproducibility.

**Questions For Authors:**

1. Potential Interaction Between Model and Data Heterogeneity: In Corollary 4.2, you focus on the effect of multi-level elastic temperature for mitigating model heterogeneity, concluding that class-specific differences are not impacted. Yet you later propose category-aware scaling to address data heterogeneity. Would class-wise temperature adjustments (Equation 4) interact with the multi-level scheme (Equation 3) in ways that actually affect model heterogeneity as well? Do you think it is necessary to analyze whether category scaling at each level might improve—or possibly interfere with—multi-level temperature adjustments?


2. Related Work on Dynamic Temperature Tuning: The authors introduce Multi-level Elastic Temperature and Category-Aware Global Temperature Scaling in the context of FL. However, there is other research in KD or general deep learning about adaptive/dynamic temperature (even if not specifically for federated settings). Could you clarify whether there are existing algorithms that similarly adjust temperature across multiple layers or categories, and if so, how your proposed approach differs? A comparison or reference to this broader literature (outside federated learning) would help situate your contributions more clearly.

**Relation To Broader Scientific Literature:**

The paper builds on and extends multiple lines of work within federated knowledge distillation and heterogeneous FL:

1. Heterogeneous FL. It aligns with prior methods such as FedMD and FedGKT in enabling cross-architecture knowledge transfer. However, whereas many of these methods rely on proxy models, public datasets, or large feature transfers, the authors focus on lightweight logit exchanges alone, making it more communication-efficient.

2. Multi-level Distillation. Inspired by hybrid distillation approaches, the paper introduces a multi-stage temperature calibration that had not previously been emphasized in the federated setting. This multi-level tactic leverages work showing intermediate-layer KD can improve model robustness and accuracy, but extends it with a dynamic temperature schedule designed for non-IID data and heterogeneity.

3. Security Extensions. The Z-score–based guard echoes recent interests in detecting malicious updates or Byzantine nodes. By combining blockchain-inspired validation with logit-based screening, ReT-FHD attempts to formalize trust boundaries and reduce risks of label-flipping or adversarial logits.

**Theoretical Claims:**

I checked the proofs for Theorem 4.1, Corollary 4.2, and Theorem 4.3 in the Appendix, and here are some observations:

**1. Boundary Conditions and Dynamic Distributions:**

Theorem 4.1 relies on taking the partial derivative of information entropy with respect to temperature under a fixed, single softmax distribution. In a multi-round FL process, however, both teachers’ and students’ logits evolve. If \tau approaches 0 or becomes very large, or if the model distribution becomes nearly deterministic at intermediate steps, the partial derivative behavior may shift drastically. The paper does not clarify how Theorem 4.1 extends to these boundary cases, nor does it show whether the distributional assumptions remain valid across rounds and layers, given that entropy typically changes along with training progress.

**2. Uniform Class Impact in Partial Derivatives:**

The proof of Corollary 4.2 concludes that all classes experience the same shift in knowledge scale by showing the partial derivatives for each class are identical. However, it does not fully justify why this equivalence holds once training proceeds and the network starts updating logits in potentially different ways for each class (especially if class distributions shift or vary across training steps). While the algebraic steps appear correct for an initial, uniformly random setup, the proof does not clarify how or whether the partial derivatives remain identical in subsequent iterations.

**3. Class-Specific Temperature and Gradient Updates:**

For Theorem 4.3, the logic that “increasing a class’s temperature increases its gradient update” is demonstrated under the corollary’s assumption of balanced classes and random initialization. This might not strictly hold in more skewed data settings or in deeper, multi-stage distillation pipelines unless further assumptions are introduced.

---

> ### Author Rebuttal · Authors · 2025-03-30
>
> We thank reviewer TEBy for the constructive comments: "…enhance accuracy under model/data heterogeneity", "…reduces communication costs compared to proxy-based methods", "… represents a novel synthesis of ideas from robust FL, KD, and malicious detection … ", "… address a variety of practical deployment scenarios". To thoroughly address your concerns, we will answer the questions one by one:
>
> **Questions:**
>
> **Q1: Potential Interaction**
> >The interaction of category temperature tuning (Eq.4) with the multi-level distillation (Eq.3) can have a **positive impact** on mitigating model heterogeneity. We facilitate knowledge transfer by applying category scaling to global logits, which simultaneously improves multi-level distillation. The ablation experiments in Tab.4 show that the combination of the two significantly improves performance, demonstrating that they are **complementary** in dealing with heterogeneity.
>
> **Q2: Differences with Existing Dynamic Temperature Methods**
> >Please refer to Reviewer JGAK's Q1 for the answer to this question.
>
> **Weaknesses:**
>
> **W1: Extension to Transformer**
> >Logits-based knowledge distillation is also **applicable to the Transformer architecture**, and similar work [1] has validated the effectiveness of hierarchical distillation on Transformers. Since we deliver logits information and eliminate architecture specificity through branching exits, our approach is more general and can be widely adapted to different architectures.
>
> **W2: More Detailed Clarity**
> >The exact model distribution for heterogeneous clients: the four models AlexNet, ShuffleNetV2, ResNet18, and GoogleNet are randomly and evenly assigned to clients.
> In distillation learning, logits serve as high-level guidance. Our elastic temperature calculation uses **next-stage logits** (vs. current-stage) to **avoid local stage overfitting and enhance stability**. Besides, the experiments on Cifar-10 can verify this:
> | Cifar-10 | next-stage | current-stage |
> | --- | --- | --- |
> | accuracy | 72.76% | 71.54% |
>
> **Responses to Questions about Theoretical Analysis**
>
> >**Theorem 4.1:** Theorem 4.1 **does not** require specific assumptions and it is valid in the framework of both multilevel elastic temperature and category-aware global temperature scaling. According to the information entropy formula, the information entropy of logits is only affected by the softmax temperature and the model output and is **only related to the temperature coefficient** when the model output **cannot be changed**. In Eq.3, we set the **initial temperature ξ** and limit its variation by a **scaling factor γ and a log function** to avoid the boundary case. In addition, in lines 237-238 of the code file client.py, we set the temperature range to **0.5 ≤ τ ≤ 5**.
> >
> >**Corollary 4.2:** We refer to **the assumptions of [4]** to simplify the theoretical proof process. In addition, Corollary 4.2 is proposed to provide theoretical support for category temperature setting. Regardless of the initial state of the model, we dynamically adjust the category temperatures of the global logits based on the local learning of different categories. Since temperature scaling is done at the softmax level, it is applied evenly to all classes in the same round, thus ensuring that the scaling effect remains consistent. In addition, we focus mainly on model heterogeneity and non-iid data distributions in FL, whereas **changes in the category distribution are not our concern**.
> >
> >**Theorem 4.3:** Although Theorem 4.3 assumes category equilibrium for analytical simplicity, temperature scaling **remains effective in heterogeneous settings**, enhancing gradient updates for difficult categories. Tab.1(a) and Fig.1 validate this in non-IID scenarios (e.g., α = 0.1, 0.5, pat), showing **significant performance gains**. Fig.2 further demonstrates that increasing distillation layers from 1 to 4 improves accuracy, plateauing at 5, confirming the validity of our multi-level mechanism.
>
> **Supplementary experiments**
> **Experiments on model heterogeneity in DeL**
> >These decentralized baselines except DeSA[3] basically **do not consider model heterogeneity**, and it is unfair to forcefully change them to baselines for heterogeneous models. Due to time constraints, we supplemented our experiments with model heterogeneity on only one dataset:
> |Flower102|DeSA|ReT-FHD|
> |---|---|---|
> |accuracy|37.17%|40.82%|
>
> **References:**
> >[1] One-for-All: Bridge the Gap Between Heterogeneous Architectures in Knowledge Distillation
> >
> >[2] Improving Adversarial Robust Fairness via Anti-Bias Soft Label Distillation
> >
> >[3] Overcoming data and model heterogeneities in decentralized federated learning via synthetic anchors
>
> Due to space constraints, I apologize for not being able to answer some of the questions, but if there are still questions, we can respond to them in the second discussion. Finally, thank you for your time and effort in reviewing our work.

---

> > ### Comment · Reviewer_TEBy · 2025-04-09
> >
> > Thanks to the authors for their response. After reading it, I would like to keep my original score.

---

### Official Review · Reviewer_JGAK · 2025-03-13

**Overall Recommendation:** 3

**Summary:**

The paper introduces ReT-FHD, which aims to improve the efficiency of federated learning by addressing model and data heterogeneity. The paper’s key motivation comes from the common weakness of current federated distillation methods, i.e., suboptimal temperature calibration during knowledge fusion. Therefore, the paper proposes several innovative mechanisms to tackle this challenge. Firstly, Multi-level Elastic Temperature is dynamic temperature adjustment mechanism to adjust the distillation intensity across model layers, which can optimize knowledge transfer between heterogeneous local models; Category-Aware Global Temperature Scaling introduces temperature calibration to individual categories, which is based on the confidence distribution in global logits and it can ensure a more personalized distillation process; and Z-Score Guard is a blockchain-verified validation mechanism to mitigate attacks Through various experiments on various datasets, the proposed framework significantly outperforms existing methods.

**Claims And Evidence:**

Yes, the authors provided the corresponding analysis and experiments for the claims.

**Essential References Not Discussed:**

Some knowledge distillation methods with dynamic temperature are not discussed.

**Experimental Designs Or Analyses:**

Yes

**Methods And Evaluation Criteria:**

Yes

**Other Comments Or Suggestions:**

1. Not all the notations are provided with the introduction.
2. Some key references are not introduced.
3. The adjustment strategy for temperature requires more detailed justifications.

**Other Strengths And Weaknesses:**

Strengths:
1.	The paper combines multi-level elastic temperature scaling, category-aware global temperature scaling, and blockchain-based security mechanisms into a single framework for federated knowledge distillation.
2.	The framework focus on reducing communication costs and enhancing security without introducing significant computational. Therefore, the application range of the framework is wider.
3.	The paper provides clear experiments across multiple benchmark datasets compared with several methods. The use of several datasets(CIFAR-10, CIFAR-100, Tiny-ImageNet, Flower102) and non-IID data distribution settings shows the comparing performance of the proposed method.  The ablation studies help prove the contribution of each component in the proposed framework.

Weaknesses
1. Dynamic temperature has been discussed by several works on knowledge distillation, but the work does not discuss much about these works, while dynamic temperature is a core contribution of the work.
2. The work seems to be a simple combination of multi-level knowledge distillation and dynamic temperature, while both multi-level knowledge distillation and dynamic temperature have been already proposed. Though the authors made several contributions by designing a Z-score based temperature adjustment and introducing it into federated learning, which indicates much novelty, I personally consider the novelty is not so sufficient.
3. The different deployment scenarios make only a little sense. Overall, the federated learning framework does not require the server to conduct much computation with privacy preservation, thus it is obvious that it can be easily to be extended to different scenarios. The deployment method on blockchain might not be proper to be considered as a main contribution.

**Questions For Authors:**

1. What is the main difference in the strategy for temperature when compared existing dynamic temperature knowledge distillation methods? Is there any specific requirements or challenges?

2. Which one contributes more to enabling the logit-based knowledge distillation to address the model heterogeneity in federated learning?  The dynamic temperature or the multi-level setting?

**Relation To Broader Scientific Literature:**

The key contribution of the paper is multi-level elastic temperature scaling.

**Theoretical Claims:**

Yes, I have checked the proofs of theorem 4.1 and corollary 4.2.

---

> ### Author Rebuttal · Authors · 2025-03-30
>
> We thank reviewer JGAK for the constructive comments: "…reducing communication costs and enhancing security …", "… provides clear experiments …". To thoroughly address your concerns, we will answer the questions one by one:
>
> **Questions:**
>
> **Q1: Differences from existing dynamic temperature knowledge distillation methods**
> >Existing dynamic temperature adjustment strategies—BGNN[1] (network-learned temperature for GNN nodes), CTKD[2] (course-learning-based scheduling), and DTKD[3] (sharpness approximation for temperature optimization)—are designed for **centralized learning** with **homogeneous architectures and IID data**, adjusting temperature **across training rounds (e.g., early vs. late epochs)**. Our approach 1) employs class-specific calibration for global logits to mitigate **non-IID label skew** and 2) assigns independent temperature coefficients to **distinct model stages (e.g., shallow vs. deep layers) within a single training round** to address **model heterogeneity**.
>
> **Q2: Which one contributes more to solving model heterogeneity in FL**
> >As detailed in **Line 32** and **Line 78**, the multi-level distillation is foundational to our heterogeneous distillation framework. As in the **Ablation Study (Tab.4)**(removing multi-level settings reduces performance by 3.08%), we validate the effectiveness in heterogeneous federated learning scenarios. Meanwhile, dynamic temperature( in **Theorem 4.1 and 4.3**) allows for more knowledge about global logits and improves performance by **2.82% (Tab.4)**.  We therefore respectfully assert that these components form an indivisible contribution core.
>
> **Weaknesses:**
>
> **W1: Not much discussion of dynamic temperature work**
> >Our motivation is to explore whether **logits** distillation can be adapted to federated heterogeneous learning. According to the answer for **Q1**, temperatures are adjusted between training rounds. This **coarse-grained adaptation** relies on global training progress and fails to localize dynamic changes within a single round. In FL, due to clients' non-IID data, **models across clients may exhibit significantly divergent optimization states within a single round**. However, cross-round temperature updates[1][2][3] require synchronization in the next round, leading to slowed convergence and even suboptimal solutions caused by delayed adaptation.
>
> **W2: The novelty is not so sufficient**
> >Our work is not a "simple" combination of multi-level distillation and dynamic temperature, as detailed in **Q1** and **W1**. We clarify our contributions as follows:
> >
> >>1. **Incorporating dynamic temperature distillation into heterogeneous FL**, enhancing knowledge transfer across diverse models;
> >>2. **Designing a Z-score-based malicious node verification mechanism**, effectively detecting and filtering abnormal logits to improve model robustness;
> >>3. **Providing a theoretical justification** for ReT-FHD’s effectiveness from the perspective of information entropy and gradient updates, offering solid theoretical support for our method;
> >>4. **Extensive experiments** on centralized FL (Tab.1 and Tab.3), decentralized FL (Tab.2), and blockchain FL (Tab.6, Tab.7, and Fig.3) validate our robustness and flexibility.
> >>
> >We respectfully request reconsideration of this core contribution.
>
> **W3: Blockchain deployment is not suitable as a main contribution**
> >We do not clarify that blockchain deployment is our core contribution.  In our contribution 3, we clarify that "our framework employs **Z-score verification** to validate logit distributions against dynamic boundaries, enabling automated reward/punishment protocols that deter malicious behaviors."We address its unique challenge in logits-sharing scenarios:
> >
> >>- **Motivation**: Traditional blockchain FL [4][5][6] validates via *model parameters*, but this fails for *logits-based distillation*.
> >>
> >>- **Contribution**: Our **Z-score validation** (Eq.7) ensures compatibility without parameter exposure, achieving:  **20.11% post-attack accuracy** (Tab.7 vs. baseline 11.43%)
> >
> >This mechanism directly enables our main contribution—**Z-Score Guard:** (Eq.7) for Standardized Finding of Malicious Logits.
>
> **References:**
> >[1]Boosting Graph Neural Networks via Adaptive Knowledge Distillation
> >
> >[2]Curriculum Temperature for Knowledge Distillation
> >
> >[3]Dynamic Temperature Knowledge Distillation
> >
> >[4]Robust blockchained federated learning with model validation and proof-of-stake inspired consensus
> >
> >[5]Blockdfl: A blockchain-based fully decentralized peer-to-peer federated learning framework
> >
> >[6]Bit-fl: Blockchain-enabled incentivized and secure federated learning framework
>
> Please let me know if I have addressed your concerns and if not, we welcome any further questions you might have. Finally, thank you for your time and effort in reviewing our work.

---

> > ### Comment · Reviewer_JGAK · 2025-04-04
> >
> > The authors have addressed some of my previous questions, and thus, I will raise the previous score to 3.

---

### Decision · Program_Chairs · 2025-05-01

**Decision:**

Accept (poster)

**Comment:**

All reviewers are in agreement that this paper makes a valuable contribution and should be accepted. While the reviewers have made some suggestions for improvement, these are generally minor and aimed at further strengthening the final version. We encourage the authors to carefully address the reviewers’ comments when preparing the camera-ready revision.